# Enhanced sodium storage in hard carbon via solvent co-intercalation electrolyte enabling Ah-level pouch cells at low temperatures

Meng Li[1], Zeping Liu[1], Yu Zhao[1], Zhaoyu Chen[2], Yu Zhang[3] ✉ & Naiqing Zhang[1] ✉

Sodium ion batteries are attracting extensive interest due to their low cost and abundant sodium resources. However, sodium ion batteries still suffer from severe performance degradation at low temperatures due to the conflict between ion desolvation and diffusion. Herein, we design a co-intercalation ether electrolyte to achieve solvent co-intercalation in the hard carbon negative electrode, thereby bypassing the slow desolvation process while ensuring rapid ion diffusion in electrolyte and hard carbon. The optimized solvation structure also promotes the formation of a thin, inorganic-rich solid electrolyte interface, facilitating interfacial ion transport. As a result, the co-intercalation electrolyte enables hard carbon to deliver good low-temperature performance, with an initial Coulombic efficiency of 80.5% at −50 °C (20 mA g$^{-1}$) and a capacity retention of 93% after 200 cycles (100 mA g$^{-1}$). Moreover, an Ah-level full cell retains cell stack-level (excluding packaging) specific energy of 163 Wh kg$^{-1}$ at 25 °C and 107 Wh kg$^{-1}$ at −50 °C (100 mA g$^{-1}$), demonstrating the practical feasibility of this strategy for wide-temperature sodium ion batteries. This work has the potential to overcome the long-standing trade-off between low-temperature ion desolvation and diffusion, offering an approach for electrolyte design toward wide-temperature sodium ion batteries.

The global energy landscape is shifting toward low-carbon solutions and electrochemical energy storage (EES) technology has become essential for supporting the expanding use of clean energy. Sodium ion batteries (SIBs) show significant potential in EES due to abundant sodium resources and cost-effectiveness[1–3]. Currently, SIBs are approaching commercialization in terms of energy density and cycle life at room temperature. However, the increasing demand for using batteries in cold environments calls for SIBs with reliable performance at low temperatures. And the most promising hard carbon (HC) negative electrode in SIBs shows unsatisfactory low-temperature performance, especially below −30 °C, due to slow ion diffusion in the electrolyte, across the solid electrolyte interface (SEI), and within HC itself[4–6]. Even worse, sluggish desolvation and thicker SEI

formation under low temperatures further hinder the ion transport at the negative electrode/electrolyte interface[7,8]. These challenges of HC under low temperatures are intrinsically connected to the electrolyte, which demands manipulation on electrolyte to ensure efficient Na$^+$ storage under low-temperature conditions.

The weakly solvated electrolytes (WSEs) are proposed as one attractive strategy in improving the low temperature performance of HC by weakening the Na$^+$-solvent interaction. Specifically, the use of low-polarity cyclic ether solvents can weaken their coordination with Na$^+$ and thus accelerate the desolvation process[9–11]. While the WSEs bring the conflict of the decrease ionic conductivity in electrolyte, because the weak Na$^+$-solvent interaction lowers the solvent's ability to dissociate sodium salts. For example, Li et al.[12] experimentally

[1]State Key Laboratory of Urban-rural Water Resource and Environment, School of Chemistry and Chemical Engineering, Harbin Institute of Technology, Harbin 150001, China. [2]Space Environment Simulation Research Infrastructure, Harbin Institute of Technology, Harbin 15006, China. [3]Department of Chemistry, Stockholm University, Stockholm 10691, Sweden. ✉e-mail: yu.zhang@su.se; zhangnq@hit.edu.cn

compared the ionic conductivity of various cyclic ether-based electrolytes and found that solvents with weaker coordination ability, while facilitating faster desolvation, resulted in electrolytes with lower ionic conductivity. And the WSE developed by Wang et al.[13] showed improved electrochemical performance at low temperatures, while its rate performance at room temperature was far inferior to that of conventional electrolytes with high ionic conductivity. Besides, these WSEs primarily focus on desolvation process with little-to-no attention to the slow ion diffusion inside HC negative electrode, and they lack validation in Ah-level cell packs under low temperatures. Hence, the design of tailor-made electrolyte to enhance the low-temperature adaptability of HC negative electrode remains challenging and is of great importance in the practical application for wide-temperature SIBs.

Herein, we design a co-intercalation electrolyte (CIE) toward HC negative electrode, successfully overcoming the trade-off between Na⁺ desolvation and ionic conductivity in electrolyte. Our CIE allows ions to bypass the slow desolvation process at interface and achieve sufficient diffusion in electrolyte and HC simultaneously. The free solvent regulation significantly widens the operating temperature of CIE to −50 °C while ensuring sufficient solvent co-intercalation for ion storage. Meanwhile, the precisely regulated solvation structure enables appropriate participation of anions in the solvation sheath, resulting in the formation of a thinner, inorganic-rich SEI. This interphase facilitates efficient Na⁺ transport across the SEI. As a result, our CIE greatly improves the low temperature performance of HC negative electrode and is further demonstrated in Ah-level SIBs cell pack. Typically, the Na||HC cell using CIE has an initial Coulombic efficiency (ICE) of 93.7% at 25 °C, and an ICE of 80.5% even at the low temperature of −50 °C, achieving a performance improvement compared to the existing electrolytes for HC. The CIE empowers the HC to deliver a capacity retention of 93.5% after 200 cycles at −50 °C. Further assembled Ah-level HC||NaNi$_{1/3}$Fe$_{1/3}$Mn$_{1/3}$O$_2$ (NFM) pouch cell can achieve specific energy of 163 Wh kg⁻¹ at 25 °C and 107 Wh kg⁻¹ at −50 °C. The CIE concept offers insights in designing electrolyte for broadening the operating temperature range of SIBs, especially under low-temperature conditions.

## Results and discussion
### Electrolyte design and solvation structure
The co-embedded behavior of the solvent can significantly accelerate ion transport by directly skipping the slow desolvation process[14–16]. This mechanism was initially discovered in graphite negative

electrodes, which enabled their application in SIBs[17,18]. However, the cycling performance was severely affected by the excessive volume expansion of graphite during charge/discharge[19]. In contrast, HC materials have a larger layer spacing and exhibit highly uniform volume expansion and contraction during charge/discharge, providing enough space for solvent embedding/de-embedding[20]. However, the behavior of solvent co-intercalation in HC has not been thoroughly investigated, especially its enhancement mechanism of ion transport rate at low temperatures has not been thoroughly explored. It has been shown that linear ether solvents such as diethylene glycol dimethyl ether (G2) can form a stable chelating coordination structure with Na⁺, which not only achieves solvent co-embedding and avoids the slow desolvation process, but also exhibits an good ability to dissociate sodium salts, thus endowing the electrolyte with high ionic conductivity[21,22]. However, when the temperature is lower than −30 °C, such linear ether-based electrolyte systems experience a sharp drop in ionic conductivity, salt precipitation and even solidification, which severely impede ion transport[23]. To solve this problem, a CIE was designed in this work. The design principle is based on the following findings: there are always some solvents in the electrolyte that do not participate in the coordination and only act as ion transport media, which are called free solvents. For this reason, we replaced this part of free solvent with 2-methyloxolane (MO) which has a lower melting point (−136 °C, Table S1) and is less polar. Through this strategy, not only the co-embeddable solvation structure is retained and the ion transport kinetics are accelerated, but also the working temperature range of electrolyte in low-temperature environment is significantly widened by increasing the solvation entropy.

To highlight the progressiveness of our designed electrolyte, we also used common co-intercalation ether electrolyte (CCE) and WSE as controls, and the specific electrolyte formulation is shown in Table S2. According to the optical photographs in Figure S1, CCE starts to have salt precipitation at −30 °C, and it is even completely solidified at −50 °C. On the other hand, WSE and our CIE remain liquid at −50 °C, which effectively guarantees the possibility of electrolyte application at low temperatures. The ion transport mechanisms of various electrolytes at low temperatures are shown in Fig. 1, among which the CIE has the fastest ion transport rate due to the presence of the co-intercalation behavior.

To investigate the solvation structure of the different electrolytes, theoretical calculations and experimental characterization were carried out. Firstly, the electrostatic potential maps and binding energies with Na⁺ of the two solvents were calculated using density functional

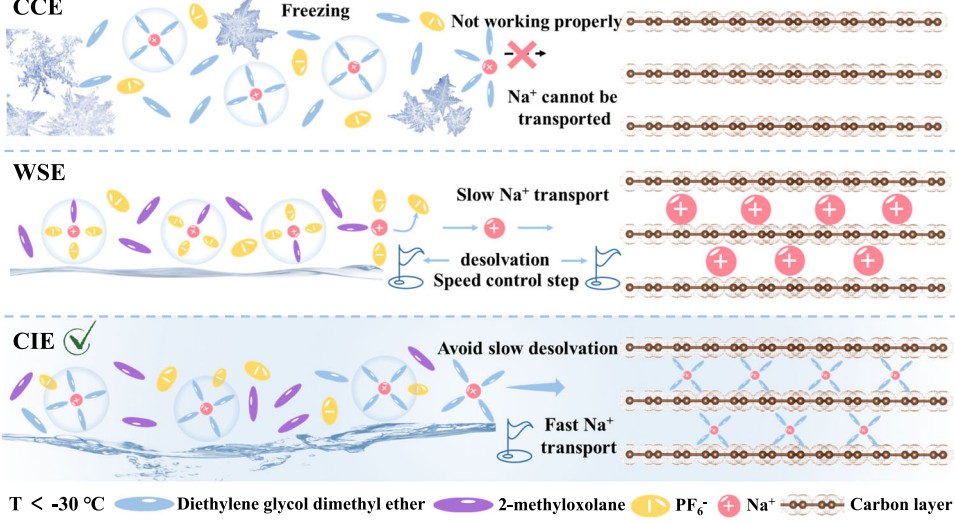

**Fig. 1 | Schematic diagram of ion transport mechanism in various electrolytes at low temperatures.**

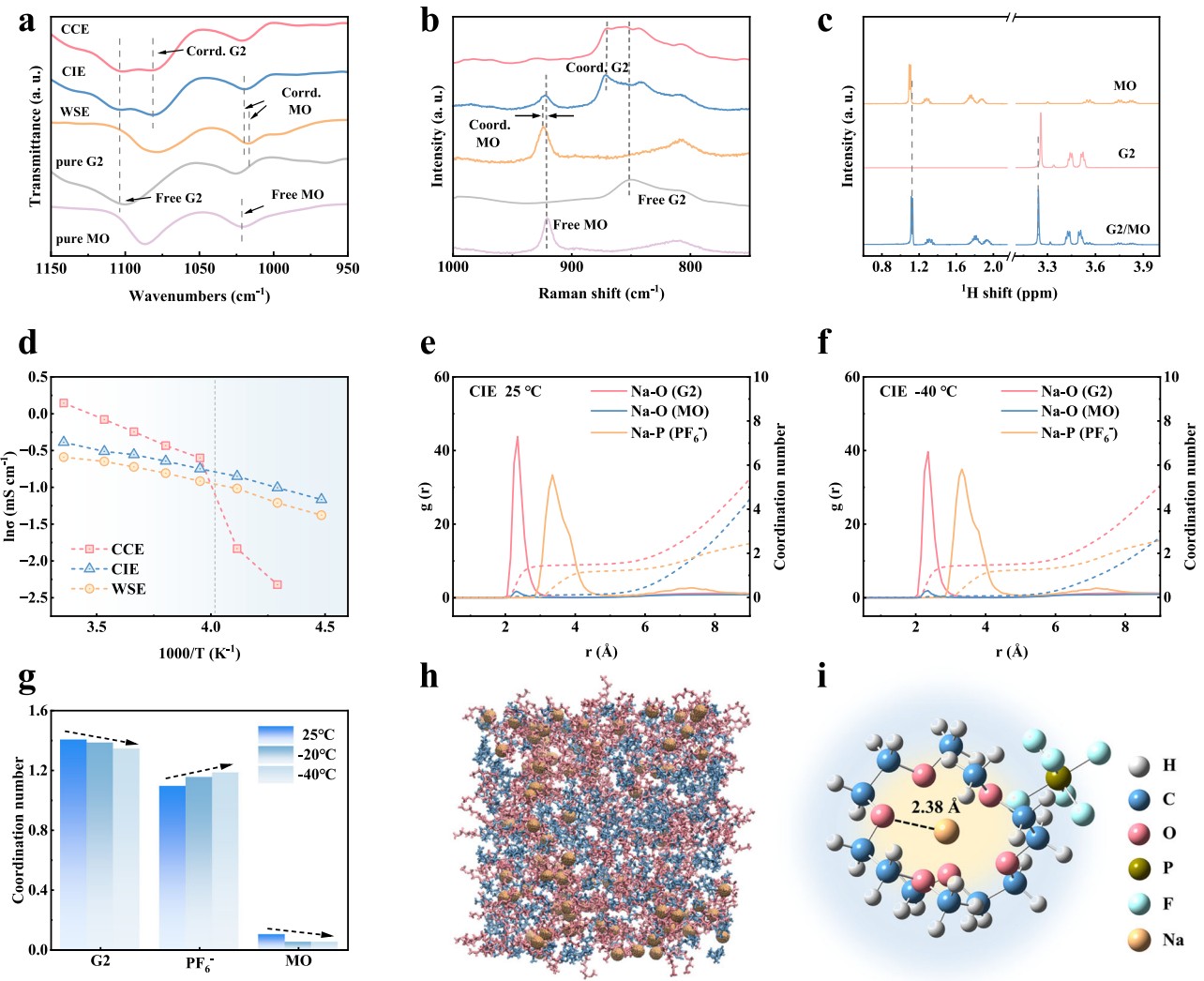

**Fig. 2 | Solvation structures of various electrolytes. a** FTIR spectra of various solvents and electrolyte. **b** Raman spectra of various solvents and electrolyte. **c** ¹H NMR spectra of MO, G2, and the G2/MO mixed solvent. **d** Ionic conductivity of various electrolytes at temperatures ranges from 25 °C to −50 °C. The RDF of CIE at (**e**) 25 °C and (**f**) −40 °C. **g** Coordination number of solvents and anions at different temperatures. **h** MD simulation snapshot of solvent environment of CIE, yellow spheres represent sodium, red represents G2 solvent, and blue represents MO solvent. **i** Typical solvated structure of CIE, white spheres represent H, blue spheres represent C, red spheres represent O, brown spheres represent P, cyan spheres represent F, and yellow spheres represent Na.

theory (DFT) (Fig. S2), which suggests that the oxygen atoms in G2 show a stronger negative electronegativity and form a chelating coordination structure with a much higher binding energy than that of MO (−0.29 eV vs. −0.13 eV). This indicates that Na⁺ will be more preferentially coordinated with G2, while MO mainly acts as a co-solvent. Fourier transform infrared spectroscopy (FTIR) was used to investigate the coordination environments in different electrolytes, as shown in Fig. 2a. The characteristic peaks at 1103 and 1021 cm⁻¹ are attributed to C-O-C stretching vibrations in pure G2 and MO solvents, respectively[24,25]. After dissolving sodium salt, a new characteristic peak appears at 1081 cm⁻¹ in CCE and CIE, which was attributed to the coordination of Na⁺ with G2 solvent, while the characteristic peak of 1016 cm⁻¹ in WSE corresponded to the coordination of Na⁺ with MO solvent. Interestingly, in CIE, the MO solvent characteristic peak remains at 1021 cm⁻¹, which indicates that MO is hardly coordinated with Na⁺. In addition, we obtain similar results by Raman (Fig. 2b), where both MO in CIE and pure MO show characteristic peak located at 920 cm⁻¹. These results suggest that in CIE, MO hardly participates in the solvation structure, and Na⁺ is mainly coordinated to G2. The existence of interaction between the solvents is further investigated by

¹H nuclear magnetic resonance (NMR), as shown in Fig. 2c. It can be found that the chemical shifts of G2 are shifted to the downfield after mixing the two solvents, which implies that there is a dipole interaction between MO and G2. This makes the density of the electron cloud around the hydrogen nucleus of G2 decrease[26]. The calculated binding energy between the MO and G2 molecules is −0.11 eV (Fig. S3), with this negative value indicating the presence of a spontaneous dipole-dipole interaction between the two. Such dipole interactions would weaken the coordination of G2 to some extent, thus providing the possibility for anions to enter the solvation structure. As shown in Figs. 2d and S4, variable-temperature conductivity tests were conducted for three electrolytes. When the temperature is lower than -20 °C, the ionic conductivity of CCE decreases rapidly, which can be attributed to the precipitation of NaPF₆ in the electrolyte. While CIE maintains a high ionic conductivity at low temperatures and is always higher than that of WSE. Therefore, compared to the WSE, the CIE designed here can effectively solve the decrease in ionic conductivity and promote ion transport at low temperatures. In addition, we also tested the physical parameters of different electrolytes, such as the electrochemical window and transference number (Figs. S5, S6).

Molecular dynamics (MD) simulations were carried out to further investigate the solvation structure of the electrolyte under different temperature gradients. The radial distribution functions (RDF) of different electrolytes are shown in Fig. 2e,f and S7-8. In CCE, the average coordination number of G2 is 1.60 and that of $PF_6^-$ is 0.87. While in CIE, the coordination number of G2 decreases to 1.41, the coordination number of $PF_6^-$ increases to 1.10, and that of MO is 0.11. This is attributed to the introduction of MO which has a dipole interaction with G2, slightly weakening the coordination of G2 and $Na^+$. In addition, MO hardly coordinates with $Na^+$, which is consistent with the results of Raman and FTIR. In WSE, the average coordination number of MO is 0.93 and $PF_6^-$ is 3.05, constituting a typical aggregate-type (AGG) solvation structure rich in anions. The solvation structure at low temperature (−20, −40 °C) was further simulated, and the changes in the coordination number of each component are compared (Figs. 2g, S9). With the decrease in temperature, the coordination of the solvent decreases, and more anions enter the solvation structure. To investigate the effect of this change on the solvation sheath, we perform a statistical analysis of the various solvent-anion coordination (Fig. S10) to accurately describe the typical solvation environments of different electrolytes at different temperatures. As shown in Fig. S11, for CCE at 25 °C and −20 °C, solvent-separated ion pairs (SSIPs) dominate. While when the temperature further decreases, the solvation structure transforms to be dominated by contact ion pairs (CIPs), suggesting that more anions will enter the solvation structure as the temperature decreases. For CIE, it always maintains the solvation sheath of the CIPs; a co-embeddable G2-$Na^+$ chelating structure is included, and a moderate amount of anions is also introduced, which facilitates the formation of anion-derived, more inorganic, and thinner SEI[27,28]. In addition, the temperature-stable solvated structure in CIE guarantees the same co-embedded behavior at low temperatures as at 25 °C. In contrast, WSE always maintains the solvation sheath of AGGs, in which more anions are introduced into the solvation structure. Too many anions can decompose rapidly at negative electrode/electrolyte interface, leading to the accumulation of inorganic substances on the surface and the formation of thicker and inhomogeneous SEI[29]. Therefore, it is necessary to introduce a moderate amount of anions like CIE to form thin SEI for effective ion transport. Figs. 2h and S12 are snapshots of MD simulations of the solvation environments of the three electrolytes. And $Na^+$ is surrounded by the G2 solvent in the CIE, ensuring the solvent co-intercalation mechanism. The solvation structures of the CIPs are further clarified by DFT, and as shown in Fig. 2i. $Na^+$ coordinated with two G2 molecules (average distance of 2.38 Å) forms the first solvation sheath, while $PF_6^-$ is mainly distributed in the second solvation sheath. Overall, we precisely regulate the solvation structure to retain the co-embeddable chelating coordination for facilitating ion transport, while also enabling an appropriate number of anions to enter the solvation sheath for promoting the reconstruction of a more favorable SEI structure.

## Electrochemical performance

To evaluate the electrochemical performance of the electrolyte, Na||HC cells were assembled. The structural characteristics of HC are shown in Fig. S13, and for HC, the reversible capacity during the first cycle is critical for less sodium loss. As shown in Fig. 3a, CIE provides a specific capacity of 300.5 mAh g$^{-1}$ and an initial Coulombic efficiency (ICE) of 93.7% during the first cycle at 25 °C, while CCE and WSE provide specific capacities of 299.6 and 300.1 mAh g$^{-1}$ and ICE of 87.7% and 91.4%, respectively. The three electrolytes deliver similar reversible capacities, while CIE has the highest ICE. The severe electrolyte decomposition creates a thick SEI and affects ICE[30,31], which may explain the lower ICE of CCE. For CIE and WSE, the introduction of anions enables the construction of inorganic-rich SEI, and the appropriate number of anions in CIE consumes less $Na^+$ during film formation, thus guaranteeing a high ICE for HC in CIE. The rate performance

of HC is shown in Figure S14, where CIE still provides a specific capacity of 195.0 mAh g$^{-1}$ at 2 A g$^{-1}$, while CCE and WSE maintain a specific capacity of 204.6 and 131.4 mAh g$^{-1}$, respectively. This phenomenon suggests that the ionic conductivity plays a crucial role in the rate performance at 25 °C. The CIE effectively improves the ionic conductivity while retaining the co-intercalation behavior, so the rate performance is close to that of CCE. While WSE has poor rate performance at 25 °C due to lower ionic conductivity. Figure 3b further shows the rate performance of HC at −20 °C. Considering that all three electrolytes deliver similar ionic conductivity at this temperature, the variability in rate performance primarily originates from the mode of ion diffusion and interface composition. Interestingly, the rate performance of WSE is better than that of CCE at low temperature, which is attributed to the formation of a thin inorganic-rich SEI. The thicker SEI of CCE restricts the rate of $Na^+$ transport and even hinders solvent co-embedding, thus resulting in poor rate performance at low temperatures. In contrast, CIE with both solvent co-embedding and thin SEI show good rate performance at low temperatures. The sodium storage performance of HC at low temperature was further tested (Fig. 3c). The HC using CIE can provide a specific capacity of 269.6 mAh g$^{-1}$ at −20 °C with an ICE of 91.4%, and maintains more than 80% of the ICE even at a low temperature of −50 °C. Such performance is comparable to existing high-performance reports (Fig. 3d). In contrast, CCE and WSE show lower capacity and ICE at low temperatures (Fig. S15). In particular, CCE can hardly work normally at temperatures below −30 °C. CIE also exhibits good cycling stability (Fig. 3e), with a capacity retention of 91.0% after 2000 cycles (capacity decay 0.0045% per cycle) at 500 mA g$^{-1}$ (25 °C). The capacity retention of CCE and WSE is 83.7% and 90.8%, respectively. Moreover, the cycling performance of the CIE at different specific currents was also tested (as shown in Fig. S16), all of which show high stability. Furthermore, the cycle stability of HC with electrolytes under low-temperature environment was tested. The capacity retention rate using CIE is as high as 96.1% after 200 cycles at −40 °C (100 mA g$^{-1}$). And the capacity retention rate is still 93.5% after 200 cycles at −50 °C, and the average Coulombic efficiency is 99.8% (Fig. 3f). Such low-temperature $Na^+$ storage performance is attributed to the enhanced overall ion transport process and anion-derived thin SEI, whereas CCE and WSE exhibit poor cycle performance at low temperatures (Fig. S17). To sum up, CIE has achieved high reversible and high-rate sodium storage at 25 °C and low temperatures with good cycling stability, which has a high potential for low-temperature SIB applications.

## Interfacial chemistry

The electrode/electrolyte interface is crucial for ion transport and cycling stability, and different electrolytes determine the structure and composition of the SEI. Firstly, the composition of SEI was characterized by X-ray photoelectron spectroscopy (XPS) (Figs. 4a-b, S18-21). Three peaks located at 284.8, 286.3, and 289.2 eV are observed in the C 1$s$ spectra, corresponding to C-C, C-O, and O-C = O, respectively[32]. This mainly corresponds to the organic components in the SEI. The higher intensity C-C peaks in the CCE indicate that the SEI contains more organic compounds. In addition, at low temperatures, the intensity of the C-C peaks of all three electrolytes is higher than that at 25 °C, which indicates that the electrolyte decomposition is more severe and more decomposition products are found in the low-temperature environment. The same results are verified in both O 1$s$ and F 1$s$ spectra, with an increase in both $Na_2CO_3$ and NaF at low temperatures. Combined with the research on solvation structure mentioned above, more anions enter the solvation sheath as the temperature decreases, and anions catalyze the formation of an SEI enriched in inorganic components[33]. WSE derived SEI contains the most inorganic compounds, especially NaF, which originates from the decomposition of $PF_6^-$ in the solvation structure. Although NaF has high mechanical strength and can effectively protect the electrode[34], excessive NaF will

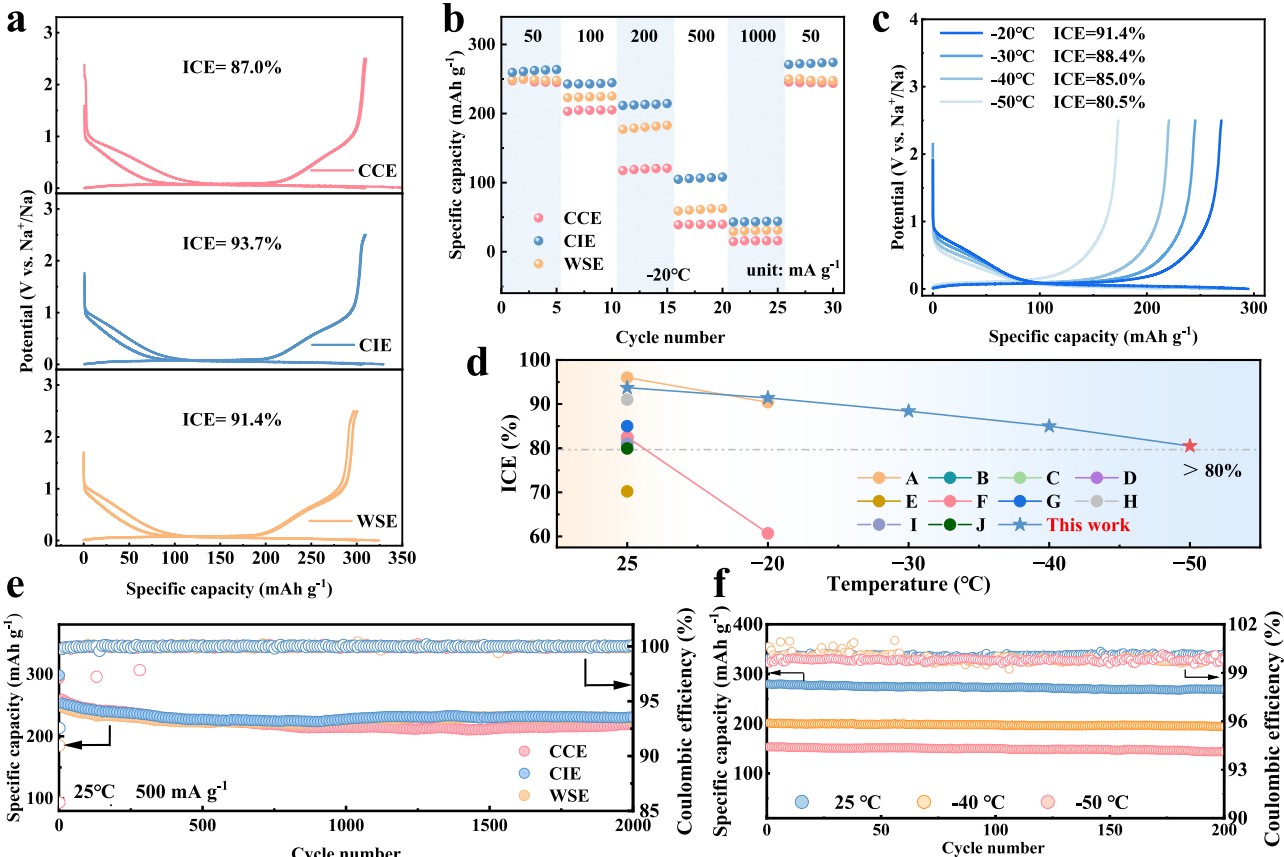

**Fig. 3 | Electrochemical performance of various electrolytes. a** Charge/discharge curves for the first three cycles of various electrolytes at 25 °C (20 mA g$^{-1}$). **b** Rate performance of various electrolytes at −20 °C. **c** Charge/discharge curves of Na || HC cells at 20 mA g$^{-1}$ in first cycle using CIE at different temperatures. **d** Comparison of the ICE at different temperatures of this work with the work reported in the literature, refer to Supplementary Table S5 for specific content. **e** Cycling performance of Na ||HC cells using various electrolytes at 25 °C (500 mA g$^{-1}$), three pre cycles of 50 mA g$^{-1}$ were conducted before the cycle. **f** Cycling performance of Na ||HC cells using CIE at 25 °C, −40 °C and −50 °C (100 mA g$^{-1}$), three pre cycles of 50 mA g$^{-1}$ were conducted before the cycle.

increase the interfacial resistance and hinder the migration of Na$^+$[35,36]. In contrast, CIE derived SEI contains an appropriate amount of NaF, which not only protects the electrode but also does not affect the rapid transport of Na$^+$. In addition, we characterized the HC electrode with different etching depths after low-temperature cycling. As etching progresses, it is found that the peak intensity of some inorganic components (such as Na$_2$CO$_3$ and NaF) in SEI decreases. This indicates that the inorganic components within the SEI are predominantly distributed in the outer layer. By comparing the variation in F element content at different etching depths (Figs. 4a-4b, and S22), CIE shows a more significant decrease in F element content after the first etching compared to WSE, and the same trend was observed for Na$_2$CO$_3$. This indicates that inorganic components of SEI in CIE are confined to a thinner surface layer. The structure of SEI was further tested by high-resolution transmission electron microscopy (HRTEM). As shown in Figs. S23-24, the HC negative electrode using CCE shows the thickest (~32 nm) and inhomogeneous SEI at 25 °C. While the SEI of WSE and CIE is thinner, especially the SEI of CIE, which is only about 16 nm. This is attributed to the participation of anions in the decomposition of CIE and WSE to form thinner SEI. At low temperatures, severe electrolyte decomposition leads to thickening of the SEI. However, even at −40 °C, the SEI of the HC negative electrode using CIE is only 30 nm (Fig. 4c), while the SEI of WSE increases to 44 nm thickness at low temperatures (Fig. 4d). Thicker SEI hinders the transport of Na$^+$, resulting in poor performance.

Time of flight secondary ion mass spectrometry (TOF-SIMS) is a characterization method capable of obtaining the 3D structure of SEI[37].

Figures S25a and 4e show the variation in the content of the main substances on the HC electrode using the CIE electrolyte with sputtering time after cycling at 25 °C and −40 °C, which is mainly composed of organic components (C$_2$HO$^-$, CH$_2$O$^-$, PO$_2^-$) and inorganic components (NaF$_2^-$, NaCO$_3^-$, NaO$^-$). The peak intensity of each component in the SEI formed at 25 °C is slightly lower than that at −40 °C, indicating that low temperature will cause more decomposition of the electrolyte, which is consistent with the XPS results. Moreover, the structure of SEI formed by WSE at −40 °C was also tested (Fig. 4f). It can be clearly observed that the peak intensity in CIE shows a sharp decrease after ~10 s of sputtering, which indicates that the electrolyte decomposition components in CIE form a thinner layer concentrated at the electrode surface. While the peak intensity of WSE decreases more gently. In addition, 3D reconstructed images visualize the depth structure of the SEI (Figs. 4g, S25b and S26), where it is evident that the WSE-formed SEI has a higher content of inorganic compounds and is evenly distributed in deeper layers of the SEI. In contrast, in CIE, both organic and inorganic compounds in the SEI are highly concentrated at the surface (especially NaCO$_3^-$ and NaO$^-$), forming a thinner and denser SEI. In conclusion, although WSE forms an inorganic-rich SEI, the excess anions in the solvation sheath cause the SEI to be too thick, which increases the interfacial resistance. Whereas, CIE is able to form an inorganic-rich, thin, and dense SEI, which effectively promotes the transport of ions across SEI (Fig. 4h)[38]. Moreover, the thinner SEI will reduce the consumption of Na$^+$, which is the reason why CIE can enable HC negative electrode to have high ICE at both room temperature and low temperatures.

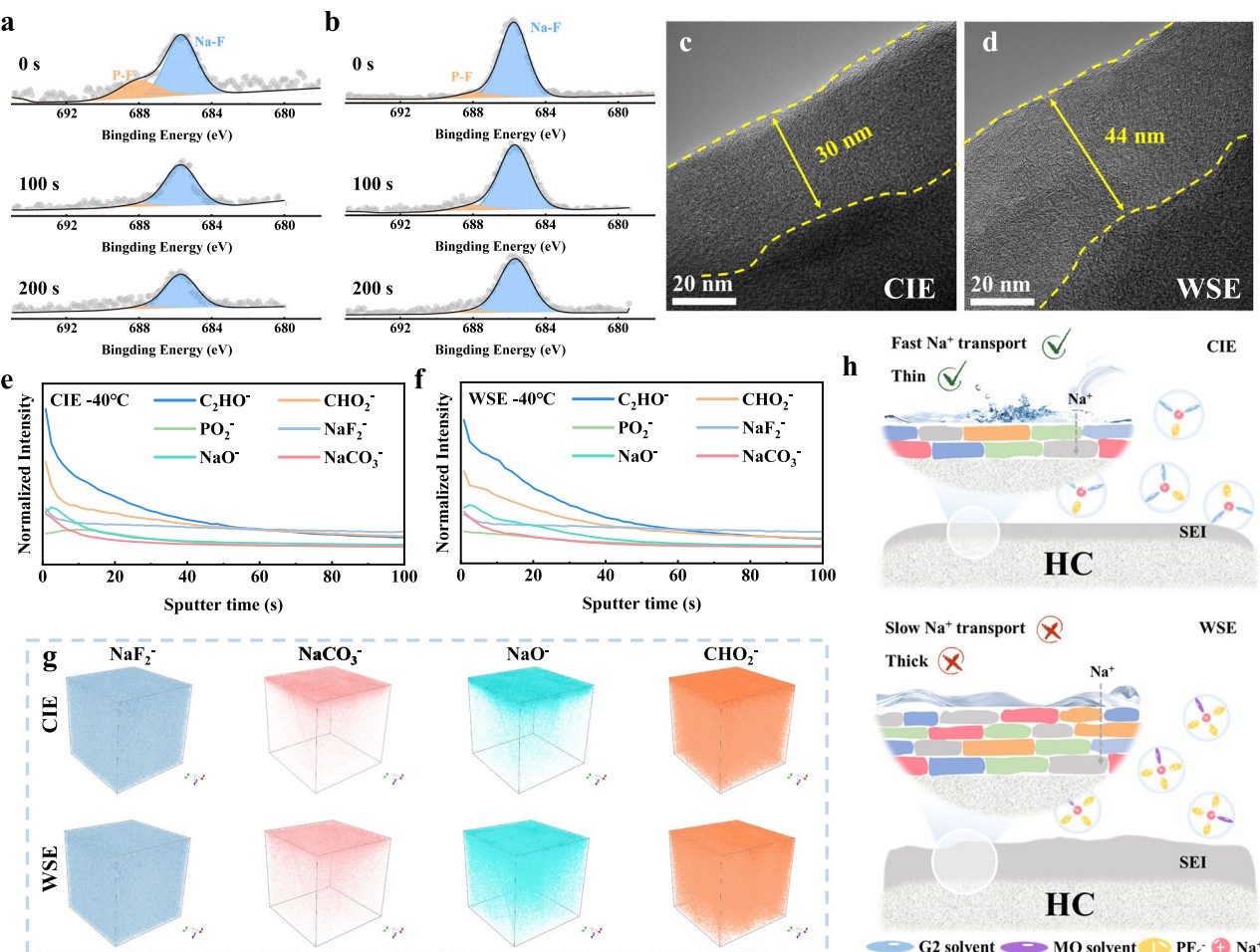

**Fig. 4 | Interface characterization of the HC negative electrode.** XPS spectra (F 1 *s*) of HC negative electrodes after 10 cycles at 50 mA g$^{-1}$ using (**a**) CIE and (**b**) WSE at −40 °C for different etching times. HRTEM images of HC negative electrodes after 10 cycles (50 mA g$^{-1}$) using (**c**) CIE and (**d**) WSE at −40 °C. Normalized TOF- SIMS depth profiles of C$_2$HO$^-$, CHO$_2^-$, PO$_2^-$, NaF$_2^-$, NaO$^-$ and NaCO$_3^-$ ionic fragments in HC negative electrode after 10 cycles (50 mA g$^{-1}$) using (**e**) CIE and (**f**) WSE at −40 °C. **g** 3D reconstruction images of NaF$_2^-$, NaCO$_3^-$, NaO$^-$ and CHO$_2^-$ resolved by TOF-SIMS in CIE and WSE. **h** Schematic illustration of the SEI formed in CIE and WSE.

## Co-intercalation behavior and ion diffusion

As confirmed in the above characterizations, the solvation structure of CIE is invariant at 25 °C and low temperatures, retaining the chelating coordination structure of G2 with Na$^+$. To confirm the co-intercalation behavior, the sodium storage process was further investigated by in situ Raman (Fig. 5a, b). There are two characteristic peaks at 1350 cm$^{-1}$ and 1580 cm$^{-1}$ corresponding to D and G peaks, respectively. As the discharge proceeds, the G peak redshifts, which is due to the insertion of Na$^+$ increasing the layer spacing and electron density, thus weakening the C-C bond[39]. When the discharge depth was further increased, the characteristic peak of the intercalation compound appeared at 1475 cm$^{-1}$ for the HC electrode in both electrolyte systems. Interestingly, when CIE is used, a new characteristic peak appeared near 1083 cm$^{-1}$, which corresponds to the Na$^+$-G2 chelate co-embedded in the carbon layer[40,41], while this peak did not appear when WSE is used. This indicates that the solvent G2 can be co-embedded in the carbon layer with Na$^+$ in CIE, whereas the solvent is completely desolvated in WSE and only Na$^+$ enters the carbon layer. To investigate the behavior of the co-intercalation in a low-temperature environment, we discharged the Na||HC cells using CIE and WSE to different potentials at −40 °C and then took out the HC electrodes for testing. To ensure credibility of the results, the electrolyte was repeatedly cleaned using solvents to eliminate residual electrolyte. As shown in Figs. 5c and S27, the G peaks are shifted compared to the raw electrodes, and the characteristic peaks of the Na-C intercalation compounds appeare,

which is the same as those observed at 25 °C. Moreover, the characteristic peak of G2 co-embedded in the carbon layer appears only when CIE is used. Figure 5d shows the XRD pattern of the HC electrode discharged to 0 V under different conditions. Compared with the raw electrode, the electrode discharged to 0 V has some small sharp peaks at 30–35° corresponding to the characteristic peaks of the quasi-metallic state Na[42], which indicates that the electrode is in a fully dis-charged state. More importantly, when discharged to 0 V, the (002) peak of HC is shifted to a lower angle. When WSE is used, only Na$^+$ is embedded in the carbon layer, and the peak shift is smaller. When CIE is used, the (002) peak shifts to a larger extent, which indicates that the solvent is co-embedded in the carbon layer with Na$^+$. In addition, when charged to 2.5 V, the (002) peak returns to the same angle as the raw electrode (due to the similarity of the XRD pattern of each group of samples after complete desodiation, only one group of samples is provided here for comparison), which indicates that the co-intercalation has high reversibility. Furthermore, we demonstrated the absence of a desolvation process by comparing liquid NMR with magic-angle spinning solid-state NMR (MAS NMR), as shown in Fig. S28. The peak of Na$^+$ in HC after using CIE is 0.9 ppm, exhibiting only a very slight shift compared to the Na$^+$ peak in the electrolyte (1.3 ppm). The slight downfield shift in the signal is attributed to the deshielding effect caused by the ring current of delocalized π electrons in the hard carbon and the paramagnetism of the interaction[43]. This indicates that during the process of Na$^+$ embedding into the

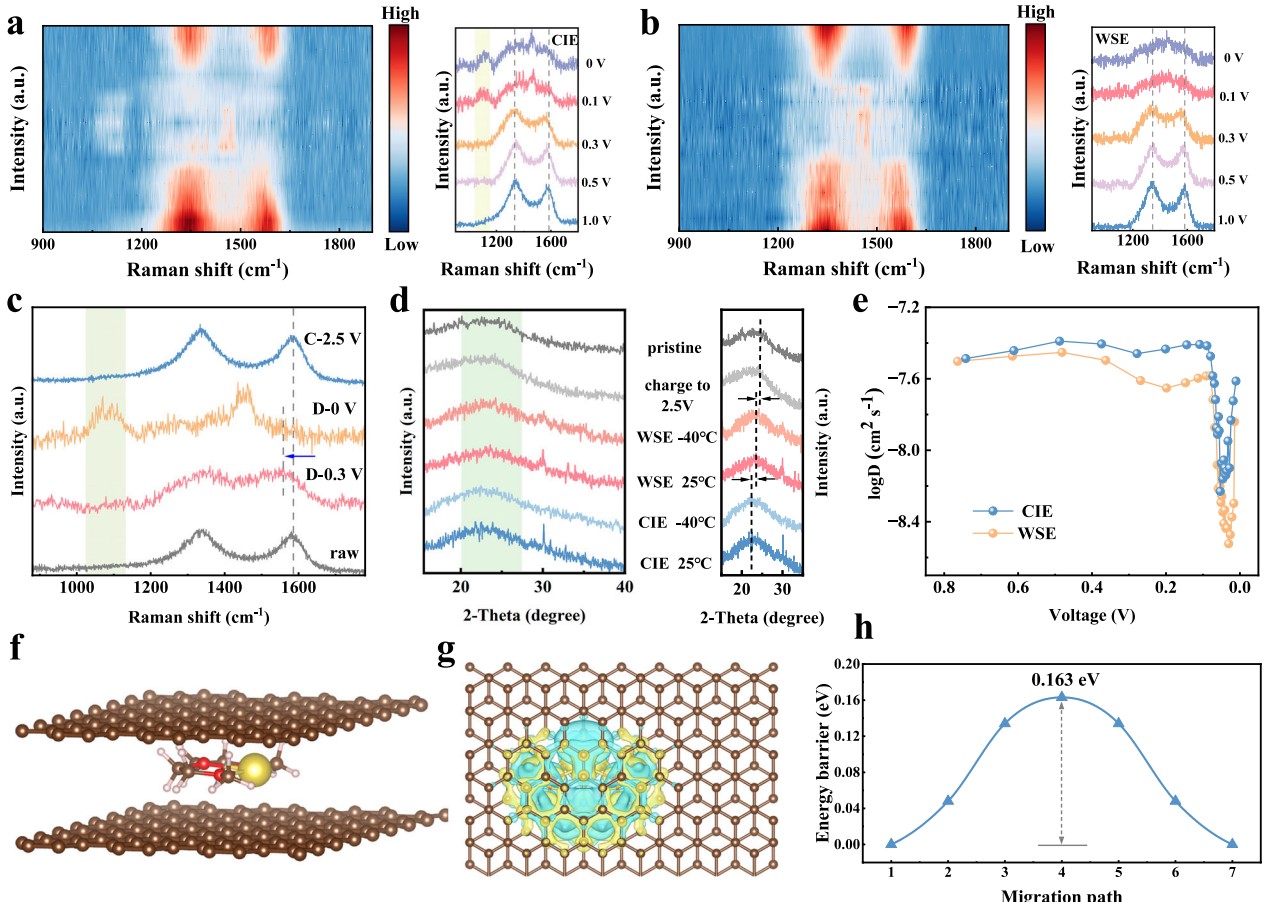

**Fig. 5 | Co-intercalation behavior and ion diffusion.** In situ Raman spectra of HC using (**a**) CIE and (**b**) WSE with a specific current of 50 mA g$^{-1}$ at 25 °C. **c** Raman spectra of HC electrodes charged/discharged at a specific current of 50 mA g$^{-1}$ to different potentials using CIE at −40 °C. **d** XRD patterns of the HC electrode after being fully discharged/charged (50 mA g$^{-1}$) in CIE and WSE. **e** Diffusion coefficients during the discharge process using different electrolytes at −40 °C. **f** Modelling of Na$^+$-G2 chelates in carbon layers, brown spheres represent C, yellow spheres represent Na, red spheres represent O, and white gases represent H. **g** Top views of charge density difference of Na$^+$-G2 in carbon layers. **h** The diffusion energy barrier.

carbon layer, its solvation environment remains intact without undergoing desolvation. However, in WSE, the MAS NMR and liquid NMR spectra exhibit significant differences, with a broad peak appearing at 8.6 ppm in the MAS NMR. This corresponds to the bare Na$^+$ into the carbon layer[44,45], indicating that WSE has undergone a complete desolvation process. We also calculated the stepwise desolvation energy via DFT for different solvation structures (Table S3)[46]. The chelation coordination structure of CIE has a desolvation energy of 25.5 kJ/mol for the first G2 and 28.3 kJ/mol for the second G2. These two values are very close and both significantly higher than the desolvation energy for removing the MO (12.3 kJ/mol). This also suggests that partial desolvation within the CIE is theoretically unlikely, enabling two G2 molecules and one Na$^+$ to embed as a whole into the carbon layer without desolvation at the interface.

The fundamental difference between desolvation and co-intercalation behavior lies in the disparity of solvation structures. For WSE, due to the weak interaction between MO solvents, the energy required to overcome the solvation energy barrier is relatively low. This enables the MO to desolvate easily and completely from the solvation sheath, with only Na$^+$ entering the interlayer, which is a highly desolvation process. For CIE, G2 exhibits strong interactions with Na$^+$, resulting in a high desolvation energy barrier that makes the desolvation process difficult. Thus, the fundamental difference between the two systems arises from the distinct interactions between the solvent and Na$^+$. In CIE, the strong solvent-ion interactions result in a high desolvation barrier, which drives the system to adopt co-intercalation.

Under electric field drive, the stable chelating solvation structure formed by G2 and Na$^+$ undergoes directed diffusion to SEI. Within the SEI, organic and inorganic components cannot achieve atomic-level compaction, inevitably leaving nanoscale voids or grain boundaries. These minute, interconnected voids provide pathways for solvated Na$^+$ clusters. Meanwhile, the CIE-formed SEI exhibits thinner layers, facilitating rapid penetration of solvated clusters through the SEI without desolvation, ultimately embedding them into the carbon layers.

In order to deeply analyze the diffusion kinetics of different electrolytes in the HC bulk phase, a series of experiments and theoretical calculations were carried out. Firstly, the ion diffusion process in HC materials at low temperature was investigated by GITT (Fig. S29). CIE has higher diffusion coefficients in both the discharge and charge processes (Figs. 5e and S31). Especially after being discharged to 0.6 V, the diffusion coefficients of CIE show an obvious gap compared with WSE, and at this time, the intercalation process starts. Higher diffusion coefficients indicate that co-intercalation favors ion transport in the bulk phase. Then, the diffusion behavior of Na$^+$ is analyzed by in situ impedance (Fig. S32). We fitted the R$_{ct}$ (charge transfer impedance) values to analyze the specific interface process (Fig. S33). The two electrolytes show a similar trend: during the initial discharge (OCV-1 V), R$_{ct}$ gradually increases, which corresponds to the polarization process. Subsequently, the increased Na$^+$ adsorbed on the active sites results in a decrease in R$_{ct}$ (1 V−0.5 V)[47]. The additional semicircle in WSE between 0.2 V−0.4 V is due to the difference in interface and ion

transport between the two electrolytes. The thickness, uniformity, and rate of ion transport of the interface may all contribute to additional impedance characteristics[48]. The 0.2–0.4 V range corresponds to the process of ion embedding between layers. For CIE, its unique co-intercalation mechanism enables high ion transport rates, with Na+ rapidly intercalating between layers and exhibiting small concentration polarization. Such rapid ion migration can cause the interfacial impedance to overlap with the charge transfer process, making it impossible to distinguish as an independent semicircle in the Nyquist plot. However, in WSE, slow ion transport and thicker SEI contribute to additional distinguishable impedance feature. As the discharge proceeds further, Na+ diffuses in the bulk phase, the electrochemical behavior gradually stabilizes, and the change of $R_{ct}$ tends to be gentle. The trend of $R_{ct}$ values during charging process is mirror image of the discharging process, which indicates that the sodium storage process in HC is reversible[49]. For CIE, this also proves the reversibility of the co-intercalation behavior. The value of $R_{ct}$ at the end of the charging process is smaller than the value at the beginning of the discharge, which is due to the formation of a stable SEI during the first charging and discharging process, thereby accelerating the ion diffusion kinetics[50]. Moreover, due to the presence of unavoidable defects in the HC material, some Na+ is irreversibly adsorbed on the HC surface, increasing the conductivity and reducing the $R_{ct}$ value. Regardless of this, CIE always has smaller $R_{ct}$ values, indicating a faster charge transfer rate. Especially during the discharge process, the $R_{ct}$ values are significantly smaller than those of WSE, which can be attributed to the higher ionic conductivity across the interface. We further investigated the diffusion kinetic mechanism of the co-intercalation behavior by DFT calculations. As shown in Fig. 5f, a chelate of Na+-G2 is introduced into the two carbon layers, and the diffusion barrier is calculated by

optimizing different migration paths (Fig. S35). Due to the electron-donating effect of oxygen in the solvent G2, the interaction of Na+ with the carbon layers is weakened (Figs. 5g, S33)[51]. Meanwhile, the co-embedding of the solvent also widens the layer spacing. The synergistic effect of these two key factors leads to a small diffusion energy barrier (0.163 eV, Fig. 5h), which is lower than that of Na+ in the interlayer (0.224 eV, Fig. S36). Such a lower diffusion energy barrier explains the merit of solvent co-intercalation mechanism.

## Electrochemical performance of pouch cell

To verify the practical applicability of CIE, we assembled a 1.2 Ah (163 Wh kg−1) pouch cell with O3-type NFM positive electrode and HC negative electrode (Fig. 6a). Figure 6b shows the cycling performance of the pouch cell at different temperatures. At 25 °C, the pouch cell provides a capacity of 1.18 Ah. As the temperature decreases, we verify the possibility of its operation at low temperatures, at −50 °C, the pouch cell still provides a capacity of 0.79 Ah (107 Wh kg−1), demonstrating a significantly better low-temperature adaptability than that of conventional electrolytes. Moreover, even at −50 °C, the charge/discharge curve maintains a stable voltage plateau (Fig. 6c). Figure 6d shows the cycling performance of the pouch cell at −20 °C with 94.1% capacity retention after 300 cycles. In addition, in order to visually verify the practical application potential, the LED lights are continuously powered at a low temperature of −50 °C. The fully-charged state pouch cells keep the LED lights working for more than 10 hours (Fig. 6e). This demonstration confirms that the CIE has good prospects for practical applications, especially at low temperatures. We further compare our results with some previously reported low-temperature sodium-ion pouch cells (Fig. 6f, Table S6)[26,52–58], and our work is comparable to most of the advanced reported works, which fully proves

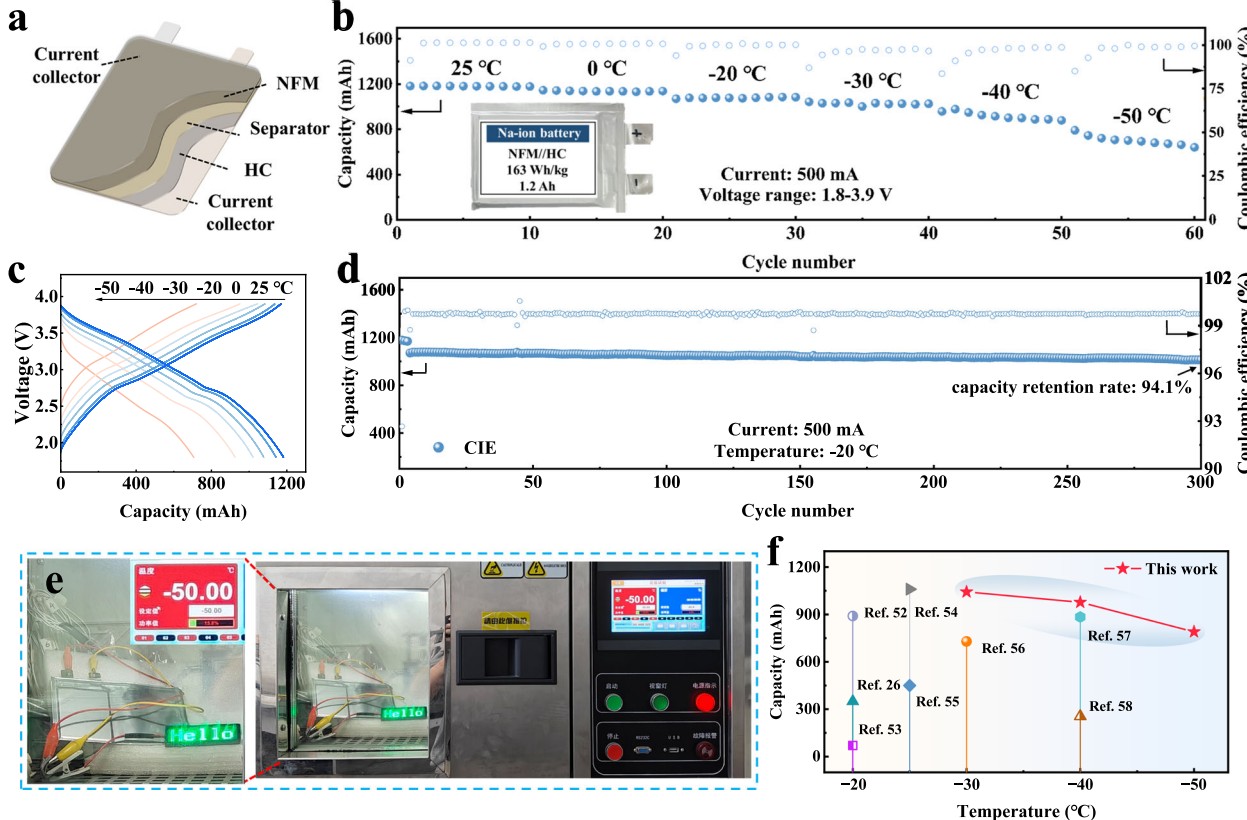

**Fig. 6 | Electrochemical performance of HC‖NFM pouch cell. a** The schematic diagram of the pouch cell. **b** Cycle performance of pouch cell at different temperatures. **c** Galvanostatic charge/discharge profiles of pouch cell at different

temperatures. **d** Cycle performance of pouch cell at −20 °C. **e** Pouch cell powers LED at −50 °C. **f** Comparison of the capacity of pouch cells at low temperatures between this work and previously reported, refer to Supplementary Table S5 for specific content.

that the CIE developed by us has important application value for SIBs in cold environments. More detailed pouch cell parameters are shown in Table S7.

In summary, we have developed a co-intercalation electrolyte based on the solvent co-embedding mechanism for efficient Na$^+$ ion storage in HC at low temperatures. The MO solvent is added into linear ether G2 to precisely regulate the solvation structure and expand the electrolyte's low-temperature range via an entropy-driven effect. Experiments and simulations reveal that the Na$^+$-G2 chelating coordination structure can be well co-embedded in HC negative electrode, avoiding the sluggish traditional desolvation process and significantly improving ion diffusion at low temperatures. Meanwhile, the high sodium salt dissociation ability of G2 ensures good ionic conductivity in electrolyte. And the dipole interaction between solvents drives an appropriate number of anions into the solvation sheath, inducing the formation of a thin layer of inorganic-rich SEI. This multi-scale synergistic design enables highly reversible and high-rate sodium storage at low temperatures. Consequently, the CIE enables HC to deliver an initial Coulombic efficiency of 80.5% at −50 °C and a capacity retention of 93% after 200 cycles. Moreover, an Ah-level full cell retains 163 Wh kg$^{-1}$ at 25 °C and 107 Wh kg$^{-1}$ at −50 °C. This study demonstrates the significance of the solvent co-embedding mechanism in improving low-temperature ion storage and provides a strategy for designing electrolytes toward cold environments.

## Methods

### Materials
The sodium hexafluorophosphate (NaPF$_6$, 99.8% min), diethylene glycol dimethyl ether (G2, 99.8% min) and 2-methyloxolane (MO, 99.8% min) were purchased from DoDo Chem. Common ether electrolyte (CCE) was prepared by dissolving 0.8 M NaPF$_6$ in G2 solvent. Weakly solvated electrolyte (WSE) was prepared by dissolving 0.8 M NaPF$_6$ in MO solvent. Co-intercalated electrolyte (CIE) was prepared by dissolving 0.8 M NaPF$_6$ in G2: MO = 7:3 vol%. All electrolyte preparation processes were carried out in a glovebox filled with Ar (H$_2$O < 0.1 ppm and O$_2$ < 0.1 ppm). The electrolyte salts and solvents we purchased are of anhydrous grade. After purchase, they are placed in the glove box for direct use.

Hard carbon (HC) materials were purchased from Baisige New Energy Co., Ltd. The HC negative electrode was prepared by mixing the active material, CMC (CMC2200, purchased from Daicel Corporation), SBR (BM-451B, purchased from Zeon Corporation) and conductive carbon black (weight ratio 92:1.5:3.5:3) in deionized water to form a homogeneous slurry. The slurry was spread on a copper foil (thickness 18 μm) and dried at 80 °C for 8–12 h. Then, it was perforated into a circular electrode sheet with a diameter of 12 mm and a loading of 2.5–3.0 mg cm$^{-2}$. Glass fiber (GF/D, Whatman) was used as separator and metallic sodium (purchased from Canrd Technology Co. Ltd.) was used as counter electrode. These components were assembled with HC negative electrode to assemble Na||HC coin cells. The diameter of the glass fiber is 16 mm and the thickness is 0.68 mm. The diameter of the sodium metal electrode is 15.6 mm and the thickness is 0.4 mm.

The Ah-level pouch batteries were composed of O3-NaNi$_{1/3}$Fe$_{1/3}$Mn$_{1/3}$O$_2$ (NFM) positive electrode and HC negative electrode coated on Al foil. The specific details and parameters are shown in Table S7.

### Electrochemical tests
The Na||HC half cells with various electrolytes were assembled in Ar-filled gloveboxes using 2032-type coin cells with Na metal and HC. The amount of electrolyte was controlled at 100 μL for each Na||HC cell. The voltage ranges of Na||HC cells and HC||NFM pouch batteries were 0 to 2.5 V and 1.8 to 3.8 V, respectively. For pouch cells, cycle at 50 mA for 2 h, followed by 100 mA for 1 h, then 200 mA for 2.5 h. After a 36 h rest period, charge at 200 mA until full capacity is reached, then discharge to 1.8 V to complete the activation process. The entire

procedure is conducted at 45 °C. The Na$^+$ storage performance was examined in a galvanostatic charge-discharge test on NEWARE Battery Test System (BTS-610, Shenzhen, China). The ion conductivity of the electrolyte was measured at different temperatures using the CHI660D electrochemical workstation through electrochemical impedance spectroscopy (EIS) technology. Galvanostatic intermittent titration technique (GITT) testing was performed by NEWARE battery system. The test was performed in the third charge-discharge cycle at 20 mA g$^{-1}$ with a pulse time for 30 min and the intervals for 2 h. Acquire 1 data point every 5 seconds. LSV (scan rate: 10 mV s$^{-1}$, voltage range: 2.5–5.5 V) tests were implemented by CHI660D electrochemical workstation. In situ EIS test: discharge from open circuit voltage to 0 V, then charge to 2.5 V, with a specific current of 20 mA g$^{-1}$. The testing frequency range is 10$^5$–10$^{-2}$ Hz, with an amplitude of 5 mV.

### Characterization
Fourier transform infrared (FTIR) spectra were performed to analyze the solvation structure on an FTIR Spectrometer (Nicolet iS10). Collect background spectra before testing and perform automatic subtraction. Raman spectra of the electrolytes were collected with a spectrometer (Thermo Fischer DXR, laser wavelength 532 nm, scanning range 700–1000 cm$^{-1}$). The SEI on HC surface was characterized by X-ray Photoelectron Spectroscopy (XPS, ThermoFisher ESCALAB 250Xi with Al Kα radiation (1486.6 eV)). The cycled HC electrode was disassembled from the tested cell in an argon-filled glove box and rinsed with solvent, and subsequently the electrode was transferred to the instrument chamber in an inert atmosphere pending XPS testing. All XPS spectra were corrected with reference to the C 1 s peak of the C-C bond located at 284.8 eV. High-resolution transmission electron microscopy (HRTEM) images were obtained by the JEM-F200 transmission electron microscope at 200 kV. The $^1$H NMR was conducted on Bruker AVANCE spectrometer (Bruker, Germany). The $^{23}$Na solid-state NMR characterization was performed on a Bruker AVANCE III 400 MHz spectrometer at a magic angle rotation (MAS) rate of 15 kHZ. The time-of-flight secondary ion mass spectrometry (TOF-SIMS) data was collected on ION-TOF-GmbH-TOF.SIMS 5-100, Germany, equipped with a 30 keV Bi$^{3+}$ gun for analysis beam, Cs$^+$ gun for sputtering beam. We normalized TOF-SIMS data using SurfaceLab software accompanying the ION-TOF instrument. The crystal structures of the HC were characterized by a powder X-ray diffractometer (X'Pert PRO, PANalytical, 5–90° within 5 min) with Cu Kα radiation ($\lambda$ = 0.15406 nm).

### Theoretical calculations
The binding energy between solvent molecules and Na$^+$ was calculated on density functional theory (DFT) by the Gaussian 16 package. The geometry optimization was conducted at B3LYP/6-31+g(d) level and a higher level of B3LYP/6-311 + +g(d,p) was applied for the single point calculation. All calculations consider the empirical revision of DFT-D3. The solvation effect was considered with the universal solvation model of polarized continuum model (PCM). The electrostatic potential involved in the analyses was evaluated by Multiwfn.

The molecular dynamics simulations were performed using the Gromacs 2019 with all-atom optimized potential liquid simulation (OPLS-AA) force field. Force field parameters for PF$_6^-$ were obtained directly from the literature, while force field parameters for other electrolyte components were generated using the LigParGen web server. Advanced restrained electrostatic potential (RESP2) atomic charges were employed, which were calculated using the Multiwfn program. The PACKMOL program was used to construct the initial simulation boxes with dimensions of 70 × 70 × 70 Å$^3$ and filled with electrolyte components. First, the temperature was allowed to slowly ramp up from 0 K to 298 K in 1 ns, then, 10 cycles of quench-anneal kinetics were performed from 298 K to 500 K within 4 ns to eliminate the persistence of the metastable state. The particle-mesh Ewald (PME) method with a cut-off distance of 12 Å was applied to treat the

electrostatic interactions and the van der Waals forces. During the pre-equilibrium simulation, the temperature is controlled by the V-rescale algorithm and the pressure is controlled by the Berendsen. Finally, MD simulations were performed in the NPT system for 20 ns at a temperature of 298 K and a pressure of 1 bar. During the final simulation, the temperature is controlled by the Nose-Hover algorithm and the pressure is controlled by the Parrinello-Rahman. The final 10-ns NPT simulation was output every 10000 steps and used to analyze the radial distribution function (g(r)) and the solvation structure. Visualization of simulated structures used the Visual Molecular Dynamics program (VMD). We used TCL script and analyzed the solvation structure based on the Python package MDTraj.

The diffusion energy barriers of $Na^+/Na^+$-G2 chelates in carbon layers were calculated using Vienna Ab-initio Simulation Package (VASP) by a method of the density-functional theory (DFT). The generalized gradient approximation (GGA), as implemented in the Perdew-Burke-Ernzerhof (PBE) functional, was employed for exchange-correlation. Dispersion effects were accounted for using the DFT-D3 method. A cutoff energy of 400 eV was applied throughout the calculations. To achieve accurate results, partial occupancy of the Kohn-Sham orbitals was allowed, utilizing the Gaussian smearing method with a width of 0.05 eV. These settings ensured energy convergence to within $10^{-5}$ eV. Atomic positions were relaxed until the forces on each atom were smaller than 0.03 eV Å$^{-1}$. Visualization was carried out using VESTA software. The atomic coordinates from calculations have been provided in the "Supplementary Data" file.

## Data availability
The data generated in this study are provided in the Source Data file. Source data are provided with this paper. The results of DFT calculation and MD simulation structure have been provided as supplementary data 1. Source data are provided with this paper.

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

## Acknowledgements
This work was supported by the National Natural Science Foundation of China (22379036), the Natural Science Foundation of Heilongjiang Province (No. JQ2021B001). Y.Z. thanks the Marie Sklodowska-Curie Actions Grant (101147049) and the Research Grant of ÅForsk Foundation (25-542). The authors would like to thank Shiyanjia Lab (from Scientific Compass www.shiyanjia.com) for providing invaluable assistance with the NMR analysis. Thanks to eseshi (www.eceshi.com) for the TOF-SIMS test. We also appreciate the Beijing Beishide Instrument for $N_2$ adsorption analyzes (BSD-660S).

## Author contributions
M.L. Y.Z. (Yu Zhang), and N.Z. conceived the idea and designed the experiments. M.L., Z.L. and Y.Z. (Yu Zhao) conducted theoretical calculations and data analysis. Z.C. characterized the materials and analyzed the data. All authors discussed the results and commented on the manuscript. M.L. wrote a draft, Y.Z. (Yu Zhang) and N.Z. revised and finalized the manuscript, and supervised the project.

## Funding

## Competing interests
The authors declare no competing interests.
