## [Transparent Peer Review file · Nature Communications]

Enhanced sodium storage in hard carbon via solvent co-intercalation electrolyte enabling Ah-level pouch cells at low temperatures

Corresponding Author: Professor Naiqing Zhang

Version 0:

Reviewer comments:

Reviewer #1

(Remarks to the Author)

Zhang et al. developed an ether-based electrolyte (CIE) for low-temperature sodium-ion batteries based on the solvent co-embedding mechanism. By introducing MO solvent to regulate the solvation structure of linear ether G2, the entropy effect is utilized to broaden the low-temperature adaptation window of the electrolyte. This collaborative design achieved reversible sodium storage at low temperature and high rate: The hard carbon anode achieved an initial efficiency of 80.5% at -50 °C, and the capacity retention rate was 93% after 200 cycles. The Ah-class full battery maintains an energy density of 107 Wh/kg at -50 °C. This work reveals the breakthrough significance of the solvent co-embedding mechanism for low-temperature ion storage. The following comments may help the author to improve the manuscript.

1. In the introduction, the author writes: "Herein, we design a co-intercalation electrolyte (CIE) toward HC anode, successfully overcoming the trade-off between Na⁺ desolvation and ionic conductivity in electrolyte. Our CIE allows ions to bypass the slow desolvation process at interface and achieve sufficient diffusion in electrolyte and HC simultaneously." Moreover, the main text also repeatedly emphasizes that the designed electrolyte only undergoes intercalation at the negative electrode and there is no desolvation behavior. However, the data in the article cannot fully support the author's core viewpoint. The fact that the original Raman and other characterizations can prove the existence of a solvent-sodium ion coordination structure in hard carbon does not indicate that the negative electrode is entirely a coordination structure and there is no desolvation process at the interface. Please supplement relevant characterization data to further support your viewpoint.

2. "As shown in Figure 3a, CIE provides a specific capacity of 300.5 mAh g⁻¹ and an initial Coulombic efficiency (ICE) of 93.7% during the first cycling at 25 °C, while CEE and WSE provided specific capacities of 299.6 and 300.1 mAh g⁻¹ and ICE of 87.7% and 91.4%, respectively." Why is the initial efficiency of the HC anode in this article so high, especially for the comparison sample CCE? In the research on the pre-sodization of HC anodes, the comparison samples mostly used 1M NaPF₆ in diethylene glycol dimethyl ether (G2), but the ICE was only 65%-70%. Similarly, 0.8 M NaPF₆ in diethylene glycol dimethyl ether (G2) was attempted, and the ICE was only 68%. Why is the ICE level so high in this article?

3. When comparing CIE and WSE electrolytes, authors should focus on discussing the underlying reasons for the differences in desolvation behavior and co-intercalation behavior of their solvation structures, rather than analyzing changes in interfacial composition and impedance. Please make relevant discussions and analyses in the text. Additionally, it is necessary to explain that at the anode interface, WSE undergoes a complete ion desolvation process, whereas CIE maintains the integrity of its solvation structure and reacts through a co-intercalation mechanism. However, why do both WSE and CIE exhibit excellent rate performance at low temperatures?

4. the author describes the formation of the interface between CIE electrolyte and HC anode as follows: "For CIE, it always maintains the solvation sheath of the CIPs, a co-embeddable G2-Na⁺ chelating structure is included, and a moderate amount of anions is also introduced, which facilitates the formation of anion-derived, more inorganic, thinner SEI. In addition, the temperature-stable solvated structure in CIE guarantees the same co-embedded behavior at low temperatures as at room temperature." The author points out that the electrolyte always maintains co-embedding behavior regardless of whether it is at room temperature or low temperature. The core function of the traditional SEI is to "allow ion conduction but prevent

solvent molecules from directly contacting the electrode active material", in order to avoid continuous solvent decomposition and electrode structure damage. The CIE strategy explicitly relies on the "overall embedding of sodium solvate ions (Na^+ - solvent clusters) into hard carbon", which means that solvent molecules will inevitably enter the electrode interior along with the ions. This is fundamentally in conflict with the "solvent barrier" function of the SEI on the surface. How to explain?

5. Recent related publications on low temperature electrolyte design are suggested to be referred, e.g., *Interdiscip. Mater.* 2023 ;2 (4): 569-588 ; *Interdiscip. Mater.* 2023; 2(6): 833-854.

Reviewer #2

(Remarks to the Author)

The topic of this manuscript is for developing low-temperature electrolyte for hard carbon anode material, which is highly relevant and timely in the field of energy storage devices for usage at low-temperature. The authors have made a significant effort to develop a new concept namely co-intercalation ether electrolyte (CIE), which helps to improve the Na^+ desolvation kinetics. The manuscript is relatively complete so after solving these issues the reviewer believes this manuscript is suitable for nature communications for publication.

1. What is the most critical parameter for formulating a CIE electrolyte? the selection of solvent or salt, or pore of hard carbon?
2. Figure 1 concluded the difference among CCE, WSE and CIE. However, I am a little confused that why WSE will not freeze and forming a dual anion solvation structure? The number of anions in electrolyte should be the same with cation.
3. Although the main idea is for the co-intercalation of electrolyte into hard carbon anode, this reviewer still would like to know the basic performance of the electrolyte such as the ionic conductivity, electrochemical stability window and sodium transference number. These physical parameters are also crucial, especially as a reference for the peer researchers.
4. In figure 5f the in-situ EIS of CIE and WSE seems different at 0.2 to 0.4 V. There is an additional semicircle in Nyquist plot of WSE. Why is this happening?
5. XPS etching is more likely to describe SEI layer on hard carbon. The authors may need to supplement two XPS etching results together with TOF-SIMS to evaluate the difference in thickness and component of SEI layer with different electrolyte.

Reviewer #3

(Remarks to the Author)

This study introduces a novel co-intercalation ether electrolyte (CIE) to overcome the long-standing trade-off between Na^+ desolvation and diffusion in hard carbon anodes for sodium-ion batteries at ultra-low temperatures. By enabling solvent co-intercalation, it bypasses sluggish desolvation and ensures rapid ion transport, leading to excellent low-temperature performance including an initial Coulombic efficiency of 80.5% at -50°C and a capacity retention of 93% after 200 cycles, even demonstrating 107 Wh kg^{-1} in Ah-level full cells at -50°C . The authors should address the following issues:

1. There are two abbreviations for one full name: CCE and CEE
2. Explain in detail why the peak of Na-P (PF_6^-) is quite large at 25 degrees Celsius in Figure 2-e, but suddenly decreases at -40 degrees Celsius. Figure 2g also has a similar problem.
3. You wrote "It can be found that the chemical shifts of G2 are shifted to the downfield after mixing of the two solvents, which implies that there is a dipole interaction between MO and G2.". I suggest that the authors calculate the interaction between MO and G2 directly via DFT.
4. It would be better to calculate the desolvation energy of the solvent more rigorously. See Table 2 of *Journal of Molecular Liquids* 395 (2024) 123817.

Version 1:

Reviewer comments:

Reviewer #1

(Remarks to the Author)

In response to the various deficiencies and uncertainties identified in this paper, the author has conducted in-depth and clear experimental characterizations for each item and supplemented them with theoretical calculations for verification. Therefore, it is proved that the CIE does not undergo desolvation at the interface and the unique ion transport mechanism in the CIE and the specific role of the SEI are elaborated in detail. This series of improvements has greatly enhanced the quality and depth of the proposed research. Thanks to these improvements, the paper now meets the criteria for publication in the journal.

Reviewer #2

(Remarks to the Author)

I have carefully checked the revised manuscript and now it meets the standard of NC.

Reviewer #3

(Remarks to the Author)

I recommend the acceptance of this version of the manuscript.

Dear Editor and Reviewers,

We sincerely thank you for processing our manuscript (Title “Solvent Co-Intercalation Electrolyte Unlocks Ah-Level Sodium Storage in Hard Carbon at Ultra-Low Temperatures”, Manuscript. ID: NCOMMS-25-46390). We appreciate the opportunity to address and clarify the issues raised in the reviewers’ report. We have carefully considered and responded to all comments to the fullest extent possible. For the ease of reference, the reviewers’ comments and suggestions are reproduced in **black**, our response is in **blue**. All changes in the revised manuscript and supporting information have been highlighted **in yellow**. We hope that the revised manuscript will be suitable for publication in this renowned journal.

Response to reviewers’ comments:

Reviewers #1	2
Reviewers #2	7
Reviewers #3	14

Response to reviewers' comments

Reviewer #1:

Zhang et al. developed an ether-based electrolyte (CIE) for low-temperature sodium-ion batteries based on the solvent co-embedding mechanism. By introducing MO solvent to regulate the solvation structure of linear ether G2, the entropy effect is utilized to broaden the low-temperature adaptation window of the electrolyte. This collaborative design achieved reversible sodium storage at low temperature and high rate: The hard carbon anode achieved an initial efficiency of 80.5% at -50 °C, and the capacity retention rate was 93% after 200 cycles. The Ah-class full battery maintains an energy density of 107 Wh/kg at -50 °C. This work reveals the breakthrough significance of the solvent co-embedding mechanism for low-temperature ion storage. The following comments may help the author to improve the manuscript.

Response: We sincerely thank Reviewers for taking the time to carefully review our manuscript. We have made further revisions to our manuscript based on your questions and suggestions. Please see below our point-by-point responses.

1. In the introduction, the author writes: “Herein, we design a co-intercalation electrolyte (CIE) toward HC anode, successfully overcoming the trade-off between Na⁺ desolvation and ionic conductivity in electrolyte. Our CIE allows ions to bypass the slow desolvation process at interface and achieve sufficient diffusion in electrolyte and HC simultaneously.” Moreover, the main text also repeatedly emphasizes that the designed electrolyte only undergoes intercalation at the negative electrode and there is no desolvation behavior. However, the data in the article cannot fully support the author's core viewpoint. The fact that the original Raman and other characterizations can prove the existence of a solvent-sodium ion coordination structure in hard carbon does not indicate that the negative electrode is entirely a coordination structure and there is no desolvation process at the interface. Please supplement relevant characterization data to further support your viewpoint.

Response: Thanks for the reviewer's valuable and insightful comment. We have carefully considered the characterization methods to demonstrate the absence of a desolvation process and referred to relevant literature. We ultimately obtained compelling evidence through a comparison of liquid and magic-angle spinning solid-state nuclear magnetic resonance (MAS NMR) data:

We sodiated HC using different electrolytes at -40 °C, then removed it and repeatedly cleaned it with solvents to eliminate the influence of residual sodium salts on the ²³Na NMR peak. The results are shown in **Figure S28**. The peak of Na⁺ in HC after using CIE is 0.9 ppm, exhibiting only a very slight shift compared to the Na⁺ peak in the electrolyte (-1.3 ppm). The slight downfield shift in the signal is attributed to the deshielding effect caused by the ring current of delocalized π electrons in the hard carbon and the paramagnetism of the interaction.¹ This indicates that during the process of Na⁺ embedding into the carbon layer, its solvation environment remains intact without undergoing desolvation. However, in WSE, the MAS NMR and liquid NMR spectra exhibit significant differences, with a broad peak appearing at 8.6 ppm in the MAS NMR. This corresponds to the bare Na⁺ into the carbon layer,^{2,3} this indicates that WSE has undergone a complete desolvation process.

Figure S28. The ^{23}Na NMR spectra of sodiated HC and electrolytes with (a) CIE and (b) WSE at $-40\text{ }^{\circ}\text{C}$.

In addition, we calculated the stepwise desolvation energies for different solvation structures, with results shown in the **Table S3**. The chelation coordination structure of CIE has a desolvation energy of 25.5 kJ/mol for the first G2 and 28.3 kJ/mol for the second G2. These two values are very close and both significantly higher than the desolvation energy for removing the MO (12.3 kJ/mol). This eliminates the possibility of partial desolvation within the CIE, enabling two G2 molecules and one Na^+ to embed as a whole into the carbon layer without desolvation at the interface.

Table S3. The desolvation energy of different solvents.

	Reaction	ΔG (kJ/mol)
$\text{Na}^+(\text{G2})_n$ ($n = 1, 2$)	$\text{Na}^+(\text{G2})_2 \rightarrow \text{Na}^+(\text{G2})_1 + \text{G2}$	25.5
	$\text{Na}^+(\text{G2})_1 \rightarrow \text{Na}^+ + \text{G2}$	28.3
$\text{Na}^+\text{-MO}$	$\text{Na}^+\text{-(MO)} \rightarrow \text{Na}^+ + \text{MO}$	12.3

In summary, *in situ* Raman spectroscopy confirms the presence of solvent in the carbon layer, while MAS NMR and theoretical calculations further confirm that no desolvation occurred at the interface: the signal characteristics of Na^+ in the solid NMR (in HC) and liquid NMR (in the electrolyte) were highly similar, indicating that the solvation structure was co-embedded as a whole in the carbon layer. Therefore, based on the above data, we believe that CIE does not undergo desolvation at the interface. Thanks again for your insightful question, which helped us improve the rigor of our paper. We have added the above content to the revised manuscript in **Page 11**.

2. “As shown in Figure 3a, CIE provides a specific capacity of 300.5 mAh g^{-1} and an initial Coulombic efficiency (ICE) of 93.7% during the first cycling at $25\text{ }^{\circ}\text{C}$, while CEE and WSE provided specific capacities of 299.6 and 300.1 mAh g^{-1} and ICE of 87.7% and 91.4%, respectively.” Why is the initial efficiency of the HC anode in this article so high, especially for the comparison sample CCE? In the research on the pre-sodization of HC anodes, the comparison samples mostly used 1M NaPF_6 in diethylene glycol dimethyl ether (G2), but the ICE was only 65%-70%. Similarly, 0.8 M NaPF_6 in diethylene glycol dimethyl ether (G2) was attempted, and the ICE was only 68%. Why is the ICE level so high in this article?

Response: We appreciate the reviewers’ valuable comments. The 87.7% ICE of the comparative

sample CEE can be explained by the following factors:

Firstly, HC materials from different sources or synthesis processes exhibit significant variations in specific surface area, pore structure, and defect concentration, which directly influence the ICE. For the study of presodiation of HC anodes, in order to highlight the modification effect, HC materials with high defect content may be used, which have high capacity in the slope region but low ICE. Some works improved ICE by reducing the specific surface area and defect concentration of HC, thereby giving it more platform capacity. For examples, when using 1M NaPF₆ in G2 as the electrolyte, the HC anode prepared by Xiong *et al.* (*Angew. Chem. Int. Ed.* 2024, 63, e202409906) achieved an ICE of 88.5%, Zhang *et al.* (*Adv. Mater.* 2024, 2412989) achieved an ICE of 90%, while that of Chou *et al.* (*Angew. Chem. Int. Ed.* 2024, 63, e202406889) reached 91.45%. These results sufficiently demonstrate that the structure of HC directly influences the ICE.

To meet practical application requirements, the HC material in our work has lower specific surface area and fewer surface defects ($I_D/I_G=1.54$), resulting in an ICE of 87.7%. More importantly, the CIE developed in this work further improves the ICE of HC to 93.7%, highlighting its advancement.

Once again, thank you for your thoughtful comments. We hope that the above explanation helps to clarify and resolve those concerns.

3. When comparing CIE and WSE electrolytes, authors should focus on discussing the underlying reasons for the differences in desolvation behavior and co-intercalation behavior of their solvation structures, rather than analyzing changes in interfacial composition and impedance. Please make relevant discussions and analyses in the text. Additionally, it is necessary to explain that at the anode interface, WSE undergoes a complete ion desolvation process, whereas CIE maintains the integrity of its solvation structure and reacts through a co-intercalation mechanism. However, why do both WSE and CIE exhibit excellent rate performance at low temperatures?

Response: Thanks for the reviewer's valuable comment.

Firstly, we delved into the fundamental causes underlying the differences in desolvation and co-intercalation behavior. The essence lies in the difference in solvation structure, which is mainly determined by the strength of the solvent-Na⁺ interaction. Based on our previous calculations of the binding energies between different solvents and Na⁺, the binding energy between the G2 molecule and Na⁺ is significantly stronger than that of the MO molecule (-0.29 eV vs. -0.13 eV). Moreover, we calculated the ΔG for complete desolvation of the solvent in the solvation shell for different electrolytes (**Table S3**). For WSE, due to the weak interaction between MO solvents, the energy required to overcome the solvation energy barrier is relatively low. This enables the MO to desolvate easily and completely from the solvated sheath, with only Na⁺ entering the interlayer, which is a highly desolvation process.

For CIE, G2 exhibits strong interactions with Na⁺. The ΔG for removing one G2 is 25.5 kJ/mol, while that for removing the second G2 is 28.3 kJ/mol. Such high desolvation energy barrier makes the desolvation process extremely difficult. Consequently, CIE adopts a unique solvent co-intercalation mechanism, in which two G2 molecules and one Na⁺ enter the carbon interlayer as an intact complex, thereby avoiding the high energy penalty of desolvation. The essence of this behavior is that the strong solvation structure tends to maintain its stability and integrity to minimize the energy barrier.

Thus, the fundamental difference between the two systems arises from the distinct interactions

between the solvent and Na^+ . In CIE, the strong solvent-ion interactions result in a high desolvation barrier, which drives the system to adopt co-intercalation.

The good rate performance of WSE at low temperatures can be attributed to following factors:

Firstly, in WSE, MO is a weakly coordinated solvent with the core advantage of extremely low desolvation energy. At low temperatures, the desolvation process is a rate controlled step of ion transport. Although WSE cannot avoid the desolvation process like CIE, the energy barrier required for weakly solvated electrolyte desolvation is relatively small, making it a common approach to improve the low-temperature performance of electrolytes (e.g., *Nat Commun.* 2023, 14, 8326; *Angew. Chem. Int. Ed.* 2024, e202400539).

In addition, WSE can form an anion-derived SEI. Although its performance is inferior to CIE, compared to other electrolytes, WSE's SEI contains more inorganic components, which effectively accelerates Na^+ transport. Consequently, WSE exhibits favorable rate performance even at low temperatures. More significantly, at low temperatures, CIE bypasses the desolvation process due to the co-intercalation mechanism, and an appropriate number of anions participate to form a thinner SEI, making its rate performance superior to WSE.

We have included the above relevant content in the revised manuscript and highlighted it in yellow.

4. the author describes the formation of the interface between CIE electrolyte and HC anode as follows: "For CIE, it always maintains the solvation sheath of the CIPs, a co-embeddable G2- Na^+ chelating structure is included, and a moderate amount of anions is also introduced, which facilitates the formation of anion-derived, more inorganic, thinner SEI. In addition, the temperature-stable solvated structure in CIE guarantees the same co-embedded behavior at low temperatures as at room temperature." The author points out that the electrolyte always maintains co-embedding behavior regardless of whether it is at room temperature or low temperature. The core function of the traditional SEI is to "allow ion conduction but prevent solvent molecules from directly contacting the electrode active material", in order to avoid continuous solvent decomposition and electrode structure damage. The CIE strategy explicitly relies on the "overall embedding of sodium solvate ions (Na^+ -solvent clusters) into hard carbon", which means that solvent molecules will inevitably enter the electrode interior along with the ions. This is fundamentally in conflict with the "solvent barrier" function of the SEI on the surface. How to explain?

Response: Thanks for the reviewer's valuable comment and question. You pointed out a contradiction between the core function of SEI (isolating solvents) and the solvent co-embedding behavior in CIE strategy. We would like to take this opportunity to clarify in detail the unique ion transport mechanism in CIE and the specific role of the SEI.

(1) Unique solvation structure

In CIE, Na^+ is tightly surrounded by G2 molecules with chelating coordination. In this structure, the solvent has a strong interaction with Na^+ , making it difficult to desolvation. Therefore, during the electric field driven transport of Na^+ to the HC anode, G2 molecules and Na^+ form stable clusters, which can be regarded as a whole and embedded together in the carbon layer along the diffusion path of Na^+ in the SEI.

(2) The objects affected by SEI's barrier function

The key to CIE design lies in introducing an appropriate number of anions to the outer layer of the solvation structure, which preferentially decompose over solvent molecules to form a thinner SEI layer with high ionic conductivity, thereby facilitating solvent co-intercalation.^{4,5} In CIE, the barrier effect of SEI still exists, but the main target is that free solvent molecules do not participate in coordination and are not driven by electric field orientation. The SEI effectively prevents these solvent molecules from directly contacting the electrode material, thereby inhibiting subsequent decomposition and further enhancing stability.

(3) Diffusion mechanism of solvent- Na^+ chelates in SEI

The stable chelating solvation structure formed by G2 and Na^+ diffuses directionally to SEI under electric field driving. Within the SEI, organic and inorganic components cannot achieve atomic-level compaction, inevitably leaving nanoscale voids or grain boundaries. These minute, interconnected voids provide pathways for solvated Na^+ clusters. Meanwhile, the CIE-formed SEI exhibits thinner layers, facilitating rapid penetration of solvated clusters through the SEI without desolvation, ultimately embedding them into the carbon layers. This diffusion phenomenon is also occurred in Xie *et al.*⁶ work for discussing the SEI formed on Nb_2O_5 anode, which lithium ions can pass through the nanoscale gaps between organic and inorganic substances in the SEI layer in a solvated form.

In summary, the CIE strategy does not deny the barrier function of the SEI, but rather focuses the barrier object of SEI from all solvents to free solvent molecules that are not driven by electric fields through a unique ion transport mechanism and SEI composition. The coordinating solvents form a stable cluster with Na^+ and pass through SEI as a whole, achieving co embedding.

Thanks again to the reviewer for raising this key question, which has prompted us to clarify the ion transport and SEI mechanism of CIE strategy more clearly. Relevant discussion has been updated in revised manuscript in Page 11.

5. Recent related publications on low temperature electrolyte design are suggested to referred, e.g., *Interdiscip. Mater.* 2023 ;2 (4): 569-588; *Interdiscip. Mater.* 2023; 2(6): 833-854.

Response: Thanks for your valuable advice. We have carefully read the references you provided and based on their insights, we have improved the content of the manuscript and cited them in the revised manuscript, ref 33 and 11.

For the first reference, a detailed review was conducted on the solvation structure of electrolytes in graphite-based lithium-ion batteries, and it was pointed out that “more CIP or AGG in the first solvation sheath can significantly improve the electrochemical performance of the system by promoting the formation of a robust anion-derived SEI.” This viewpoint provides important theoretical support for us to explain why the SEI film structure formed by CIE at low temperatures is more stable. Therefore, we cited this literature in the relevant section of the manuscript discussing the stability of low-temperature SEI (ref. 33). The second reference emphasizes the core role of solvation structures in regulating electrode/electrolyte interface reactions, and delves into the interactions between cations, anions, and solvent molecules. In addition, the review also points out that solvation structures and their induced interface characteristics have a decisive impact on rechargeable batteries. This viewpoint is highly consistent with the focus of our work, so we have cited this literature (ref. 11) in the introduction to better explain the research background and motivation.

Response to reviewers' comments

Reviewer #2:

The topic of this manuscript is for developing low-temperature electrolyte for hard carbon anode material, which is highly relevant and timely in the field of energy storage devices for usage at low-temperature. The authors have made a significant effort to develop a new concept namely co-intercalation ether electrolyte (CIE), which helps to improve the Na⁺ desolvation kinetics. The manuscript is relatively complete so after solving these issues the reviewer believes this manuscript is suitable for nature communications for publication.

Response: We sincerely thank Reviewer's valuable and positive comments on our manuscript. We have made further revisions to our manuscript based on Reviewer's valuable suggestions. We believe that the additional supplements and the corresponding revisions to address the comments substantially strengthen our manuscript. Please see below our point-by point responses.

1. What is the most critical parameter for formulating a CIE electrolyte? the selection of solvent or salt, or pore of hard carbon?

Response: Thanks for the reviewer's valuable question. The selection of solvent to be the most critical parameter in preparing CIE. This is because the solvent directly determines the formation of the solvation structure, thereby governing whether co-intercalation behavior occurs and its reversibility.

Firstly, strong coordinating solvents are indispensable. For example, G2 can form strong interactions with Na⁺ through chelating coordination, resulting in a high desolvation energy barrier. This ensures that solvent molecules can co-embed with Na⁺ into HC. Secondly, to achieve co-intercalation behavior at low temperatures, it is also necessary to introduce another low-melting-point solvent to replace the non-coordinated free solvent within the solvation structure. This solvent requires weak interactions with Na⁺ to prevent participation in the first solvation shell, while its low melting point broadens the liquid phase of the electrolyte at low temperatures. Moreover, such weakly coordinated solvents can introduce an appropriate quantity of anions into the outer solvation shell through dipole-dipole interactions with strongly coordinated solvents. This promotes the preferential decomposition of anions to form a thin, stable SEI. Therefore, the properties of the solvent fundamentally determine the solvation structure and co embedding behavior, which are the most critical influencing factors.

For sodium salts, we should select those with high ionic conductivity and easy dissociation. Altering the salt type or concentration mainly influences the kinetic transport and interfacial composition of the electrolyte, thereby affecting the electrochemical performance. However, these factors play an optimization role more on the basis of the occurrence of the co-intercalation mechanism, rather than determining whether co-intercalation itself can occur. Therefore, although the selection of sodium salt is very important, it is still a secondary parameter.

As for the pore structure of hard carbon, it may exert a certain influence on the initial coulombic efficiency or electrolyte wettability, but these are secondary effects that occur under the premise of co-

intercalation mechanism. The pore structure itself does not directly participate in or determine the stability of solvation structure and co-embedded clusters, so its influence is relatively limited.

2. Figure 1 concluded the difference among CCE, WSE and CIE. However, I am a little confused that why WSE will not freeze and forming a dual anion solvation structure? The number of anions in electrolyte should be the same with cation.

Response: Thanks a lot for the reviewer's careful reading and insightful comment.

WSE remains liquid primarily because MO has a low melting point (-136 °C), enabling WSE to maintain a wide liquid range at low temperatures. Regarding the apparent formation of a multi-anion solvation structure in WSE, which seems to contradict the principle of electrical neutrality, arises from a schematic representation. Figure 1 depicts the most typical solvation structure in WSE as derived from molecular dynamics (MD) simulations.

The MD simulation quantifies the number of neighboring particles around Na^+ over a period of motion. The coordination number represents the statistical average of anions and solvent molecules at varying distances from Na^+ (*Nat Energy*. 2024, 9, 57-69.; *Nat Commun*. 2025, 16, 7917; *Adv. Mater*. 2023, 35, 2208340). For WSE, we calculated a PF_6^- anion coordination number of 3.05 and a MO solvent of 0.93. In addition, we have calculated the frequency of various coordination situations in WSE, and $[\text{Na}^+\text{-anion}_3\text{-solvent}_1]$ has the highest frequency, representing the most typical solvation structure. Therefore, we have plotted the most typical first solvation structure of WSE in Figure 1, which may have caused the confusion.

Your belief that the electrolyte is generally electrically neutral is absolutely correct. Although the most typical solvation structure of WSE is a coordination of one Na^+ and three anions, this does not mean that all Na^+ strictly maintain this coordination structure at all times. And, in our statistics, we focus on a single Na^+ as the center, but for a Na^+ coordinated by three anions will also have these three anions shared and coordinated by other cations. In other words, an anion does not only belong to a Na^+ , it may be present in the solvent sheath of two or more cations simultaneously. To provide you with a clear visual understanding, we have drawn a schematic diagram (**Figure R2.1**). This sharing mechanism ensures the presence of rich anions in the solvation structure of a single Na^+ , but the number of anions and cations in the entire electrolyte system is strictly equal. This explains why weakly solvation electrolytes form anion-rich coordination (*Nat Commun*. 2025, 16, 3344; *Angew. Chem. Int. Ed*. 2025, 64, e202508152).

Figure R2.1. Schematic diagram of coordination in the anion-rich solvation structure.

In summary, Figure 1 is intended to illustrate the typical solvation structure of a single sodium ion,

which may have caused some confusion. We sincerely appreciate your insightful question, which has allowed us to clarify this point more clearly. We hope the above explanation satisfactorily addresses your concerns.

3. Although the main idea is for the co-intercalation of electrolyte into hard carbon anode, this reviewer still would like to know the basic performance of the electrolyte such as the ionic conductivity, electrochemical stability window and sodium transference number. These physical parameters are also crucial, especially as a reference for the peer researchers.

Response: Thanks for the reviewer's valuable comment and suggestion. Following the suggestion, we have supplemented the corresponding experiments and conducted systematic testing on the fundamental physicochemical parameters of the electrolyte including ionic conductivity, electrochemical stability, and transference number).

The ionic conductivity of the electrolyte was tested using the AC impedance method (**Figure S4 and 2d**). When the temperature is below $-20\text{ }^{\circ}\text{C}$, the ionic conductivity of CCE declines rapidly, whereas CIE maintains a high ionic conductivity at low temperatures, consistently higher than WSE. Therefore, compared to WSE and CCE, our designed CIE effectively addresses the issue of low ionic conductivity at low temperatures, thereby promoting ion transport.

Figure S4 and 2d. Nyquist plots of (a) CCE, (b) CIE and (c) WSE at different temperatures. (d) Ionic conductivity of various electrolytes at temperatures ranges from $25\text{ }^{\circ}\text{C}$ to $-50\text{ }^{\circ}\text{C}$.

We assembled $\text{Na}||$ stainless steel cells and evaluated the electrochemical stability window of the electrolyte using linear sweep voltammetry (LSV). The results are shown in **Figure S5**, where the oxidation decomposition potential for WSE is approximately 3.7 V, CIE is 4.2 V, and CCE is 4.3 V. For SIBs, the typical operating voltage for the cathode is around 4 V, and the electrochemical stability window at 4.2 V for CIE ensures that the electrolyte works well within the normal operating voltage

range.

Figure S5. LSV curves of different electrolytes.

We measured the transference numbers of different electrolytes by combining chronoamperometry curves with AC impedance (**Figure S6**). Tests were conducted in Na||Na symmetric cells, with transference numbers calculated using the following formula:

$$t_+ = \frac{I_s(\Delta V - I_0 R_0)}{I_0(\Delta V - I_s R_s)}$$

Where I_0 is the initial current, I_s is the steady-state current, R_0 is the pre-polarization impedance, R_s is the post-polarization impedance, and ΔV is the DC bias voltage (10 mV). The calculated t_+ for CCE, CIE and WSE were 0.52, 0.61 and 0.66 respectively. The anion-dominated solvation structure contributes to enhanced ionic transference number, hence WSE has the highest t_+ . Due to the introduction of MO, anions are introduced into the outer layer of the solvation sheath, resulting in a higher ion transfer number of CIE than CCE, which is beneficial for reducing polarization and improving rate performance.

Figure S6. t_+ and corresponding chronoamperometry profiles of Na||Na symmetric cells with (a) CCE, (b) WSE and (c) CIE.

4. In figure 5f the in-situ EIS of CIE and WSE seems different at 0.2 to 0.4 V. There is an additional

semicircle in Nyquist plot of WSE. Why is this happening?

Response: Thanks for the reviewer's valuable question. The additional semicircle in the Nyquist plot of WSE is due to the differences in the interface and ion transport between the two electrolytes. The thickness, uniformity, and rate of ion transport of the interface may all contribute to additional impedance characteristics.⁷ The 0.2-0.4 V range corresponds to the process of ion embedding between layers. For CIE, its unique co-intercalation mechanism enables high ion transport rates, with Na⁺ rapidly intercalating between layers and exhibiting small concentration polarization. Such rapid ion migration can cause the interfacial impedance to overlap with the charge transfer process, making it difficult to distinguish as an independent semicircle in the Nyquist plot. However, Na⁺ insertion in WSE requires a complete desolvation process. Due to the presence of more anions in the solvation structure of WSE, it will decompose in a short period of time and rapidly accumulate on the negative electrode to form SEI, which is thicker.⁸ Therefore, in WSE, ion transport is a relatively slow step, thereby contributing an additional, distinguishable impedance feature (an additional semicircle). As the sodiation process progresses (<0.2 V), the interlayer Na⁺ concentration gradually saturates, concentration polarization decreases, and the additional semicircle no longer prominent. **The relevant discussion has been supplemented in Page 12 in revised manuscript.**

5. XPS etching is more likely to describe SEI layer on hard carbon. The authors may need to supplement two XPS etching results together with TOF-SIMS to evaluate the difference in thickness and component of SEI layer with different electrolyte.

Response: Thanks a lot for the reviewer's careful reading and insightful comment. Following the suggestion, we have supplemented the XPS depth etching data for HC electrodes after cycling at low temperatures for different electrolytes (**Figures S19-21**).

As etching progresses, it was found that the peak intensity of some inorganic components (such as Na₂CO₃ and NaF) in SEI decreased. This indicates that the inorganic components within the SEI are predominantly distributed in the outer layer. By comparing the variation in F element content at different etching depths (**Figure S22**), we observe that CCE consistently exhibits lower F content. In contrast, WSE forms a F-rich SEI due to the presence of abundant anions in the solvation structure, but this may increase interfacial resistance and hinder Na⁺ migration. Another noteworthy point is that CIE shows a more significant decrease in F element content after the first etching compared to WSE, and the same trend was observed for Na₂CO₃. This further indicates that SEI components in CIE are more concentrated at the surface, which is conducive to the formation of thinner SEI. This finding is consistent with TOF-SIMS results.

Figure S19. XPS spectra of cyclized HC anode after different etching times using CCE at -20 °C.

Figure S20 and 4a. XPS spectra of cyclized HC anode after different etching times using CIE at -40 °C.

Figure S21 and 4b. XPS spectra of cycled HC anode after different etching times using WSE at -40 °C.

Figure S22. The variation of F element content in SEI layer under different etching times.

We believe that these supplementary experiments have helped us to analyze the differences in thickness and composition of SEI more deeply. We would like to reiterate our gratitude again for taking the time and effort to review our manuscript. The above content has been added to the revised manuscript.

Response to reviewers' comments

Reviewer #3:

This study introduces a novel co-intercalation ether electrolyte (CIE) to overcome the long-standing trade-off between Na⁺ desolvation and diffusion in hard carbon anodes for sodium-ion batteries at ultra-low temperatures. By enabling solvent co-intercalation, it bypasses sluggish desolvation and ensures rapid ion transport, leading to excellent low-temperature performance including an initial Coulombic efficiency of 80.5% at -50°C and a capacity retention of 93% after 200 cycles, even demonstrating 107 Wh kg⁻¹ in Ah-level full cells at -50°C. The authors should address the following issues:

1. There are two abbreviations for one full name: CCE and CEE

Response: Thanks to the reviewer for astutely pointing out the issue of inconsistent electrolyte abbreviations. We have thoroughly reviewed the entire manuscript and uniformly corrected all references to the electrolyte abbreviation to CCE.

2. Explain in detail why the peak of Na-P (PF₆⁻) is quite large at 25 degrees Celsius in Figure 2-e, but suddenly decreases at -40 degrees Celsius. Figure 2g also has a similar problem.

Response: Thanks for the reviewer's valuable comment. We have carefully examined the issue you raised and found that it is due to color labeling errors in the data of Na-P (PF₆⁻) and Na-O (MO) in Figure 2f (-40 °C) of our manuscript, which caused you to have such doubts. We offer our sincere apologies for this oversight in our manuscript, which has now been rectified.

In fact, Na-P (PF₆⁻) corresponds to the yellow data line in Figure 2f. When the temperature decreases from 25°C to -40°C, the peak intensity of Na-P (PF₆⁻) actually increases slightly. This indicates that as the temperature decreases, the anion's coordination tendency increases, leading to an increase in the coordination number. This result is consistent with that shown in Figure 2g.

Figure 2e, f. The RDF of CIE at (e) 25°C and (f) -40°C.

Thanks for raising this issue, which has enabled us to identify this important drawing error. We have redrawn Figures 2e-f in the manuscript to ensure that the color labeling of all data lines is accurate and correct.

3. You wrote "It can be found that the chemical shifts of G2 are shifted to the downfield after mixing of the two solvents, which implies that there is a dipole interaction between MO and G2.". I suggest that the authors calculate the interaction between MO and G2 directly via DFT.

Response: Thanks for the reviewer's valuable comment and suggestion. We have calculated the interaction between MO and G2 *via* DFT, using the same basis set for all calculations (see supporting information). The calculated binding energy between the MO and G2 molecules is -0.11 eV (**Figure S3**), this negative value indicating the presence of a spontaneous dipole-dipole interaction between the two.

$$E_b = -0.11 \text{ eV}$$

Figure S3. The binding energy between the MO and G2.

4. It would be better to calculate the desolvation energy of the solvent more rigorously. See Table 2 of Journal of Molecular Liquids 395 (2024) 123817.

Response: Thanks for the reviewer's valuable comment and suggestion. We have carefully read the literature you recommended and calculated the desolvation energies of different solvents with reference to Table 2.

The results are shown in **Table S3**. The chelation coordination structure formed by G2 and Na^+ has a desolvation energy of 25.5 kJ/mol when the first G2 molecule is removed, and the energy required to remove the second G2 molecule is 28.3 kJ/mol, both of which are much higher than the 12.3 kJ/mol required to remove the MO solvent. This indicates that desolvation is difficult to occur in CIE, thereby enabling the possibility of solvent co-embedding. We have cited this reference in the corresponding section of the revised manuscript (ref. 46).

Table S3. The desolvation energy of different solvents.

	Reaction	ΔG (kJ/mol)
$\text{Na}^+(\text{G2})_n$ ($n = 1, 2$)	$\text{Na}^+(\text{G2})_2 \rightarrow \text{Na}^+(\text{G2})_1 + \text{G2}$	25.5
	$\text{Na}^+(\text{G2})_1 \rightarrow \text{Na}^+ + \text{G2}$	28.3
$\text{Na}^+\text{-MO}$	$\text{Na}^+(\text{MO}) \rightarrow \text{Na}^+ + \text{MO}$	12.3

The literature provided by the reviewers has offered valuable guidance, enabling us to enhance the rigor of our desolvation energy calculations. We have added the calculation results and discussions to the revised manuscript, hoping that the above modifications can meet your requirements.

References:

- 1 Chen, Y. *et al.* Ultrafast Cointercalation Chemistry for Low-Temperature Sodium-Ion Batteries. *Journal of the American Chemical Society* **147**, 20431-20441 (2025). <https://doi.org/10.1021/jacs.5c02229>
- 2 Gotoh, K. *et al.* NMR study for electrochemically inserted Na in hard carbon electrode of sodium ion battery. *Journal of Power Sources* **225**, 137-140 (2013). <https://doi.org/https://doi.org/10.1016/j.jpowsour.2012.10.025>
- 3 Kim, J.-B. *et al.* Microstructural Investigation into Na-Ion Storage Behaviors of Cellulose-Based Hard Carbons for Na-Ion Batteries. *The Journal of Physical Chemistry C* **125**, 14559-14566 (2021). <https://doi.org/10.1021/acs.jpcc.1c03984>
- 4 Yang, Y. *et al.* Rechargeable LiNi_{0.65}Co_{0.15}Mn_{0.2}O₂||Graphite Batteries Operating at -60 °C. *Angewandte Chemie International Edition* **61**, e202209619 (2022). <https://doi.org/https://doi.org/10.1002/anie.202209619>
- 5 Dong, R. *et al.* Elucidating the Mechanism of Fast Na Storage Kinetics in Ether Electrolytes for Hard Carbon Anodes. *Advanced Materials* **33**, 2008810 (2021). <https://doi.org/https://doi.org/10.1002/adma.202008810>
- 6 Yu, Y. *et al.* Kinetic pathways of fast lithium transport in solid electrolyte interphases with discrete inorganic components. *Energy & Environmental Science* **16**, 5904-5915 (2023). <https://doi.org/10.1039/D3EE02048G>
- 7 Huang, R. W. J. M., Chung, F. & Kelder, E. M. Impedance Simulation of a Li-Ion Battery with Porous Electrodes and Spherical Li⁺ Intercalation Particles. *Journal of The Electrochemical Society* **153**, A1459 (2006). <https://doi.org/10.1149/1.2203947>
- 8 Li, X. *et al.* Fast Interfacial Defluorination Kinetics Enables Stable Cycling of Low-Temperature Lithium Metal Batteries. *Journal of the American Chemical Society* **146**, 17023-17031 (2024). <https://doi.org/10.1021/jacs.3c14667>